# WRN helicase safeguards deprotected replication forks in *BRCA2*-mutated cancer cells

Arindam Datta[1], Kajal Biswas[2], Joshua A. Sommers [1], Haley Thompson[1], Sanket Awate[1], Claudia M. Nicolae [3], Tanay Thakar [3], George-Lucian Moldovan [3], Robert H. Shoemaker[4], Shyam K. Sharan [2] & Robert M. Brosh Jr [1✉]

The tumor suppressor BRCA2 protects stalled forks from degradation to maintain genome stability. However, the molecular mechanism(s) whereby unprotected forks are stabilized remains to be fully characterized. Here, we demonstrate that WRN helicase ensures efficient restart and limits excessive degradation of stalled forks in BRCA2-deficient cancer cells. In vitro, WRN ATPase/helicase catalyzes fork restoration and curtails MRE11 nuclease activity on regressed forks. We show that WRN helicase inhibitor traps WRN on chromatin leading to rapid fork stalling and nucleolytic degradation of unprotected forks by MRE11, resulting in MUS81-dependent double-strand breaks, elevated non-homologous end-joining and chromosomal instability. WRN helicase inhibition reduces viability of BRCA2-deficient cells and potentiates cytotoxicity of a poly (ADP)ribose polymerase (PARP) inhibitor. Furthermore, BRCA2-deficient xenograft tumors in mice exhibited increased DNA damage and growth inhibition when treated with WRN helicase inhibitor. This work provides mechanistic insight into stalled fork stabilization by WRN helicase when BRCA2 is deficient.

[1] Translational Gerontology Branch, National Institute on Aging, NIH, Baltimore, MD 21224, USA. [2] Mouse Cancer Genetics Program, Center for Cancer Research, National Cancer Institute, NIH, Frederick, MD 21702, USA. [3] Department of Biochemistry and Molecular Biology, The Pennsylvania State University College of Medicine, Hershey, PA 17033, USA. [4] Chemopreventive Agent Development Research Group, Division of Cancer Prevention, National Cancer Institute, NIH, Rockville, MD 20850, USA. ✉email: broshr@mail.nih.gov

Germline mutations in *BRCA1* and *BRCA2* account for the majority of hereditary breast and ovarian cancers[1]. In addition to defective double-strand break (DSB) repair[2], compromised fork stability significantly contributes to heightened genomic instability and chemosensitivity in *BRCA*-deficient tumors[3,4]. BRCA2 safeguards genome integrity by protecting stalled replication forks from unscheduled nucleolytic degradation in human cells[4,5]. However, the duty of fork protection extends beyond BRCA1/2 and solicits the involvement of other stress response proteins including ubiquitinated-PCNA[6], RAD51 paralogs[7], FANCD2[8], FANCA[8], FANCJ[9], BODL1[10], CTIP[11], Abro1[12], WRNIP1[13], EXD2[14], RIF1[15], and RAD52[16]. These fork stabilizing factors are unique and operate independently or synergistically with BRCA1 and BRCA2 to maintain fork integrity[11,14,17]. Understanding how fork processing proteins functionally collaborate during replication stress to stabilize forks has important implications for genomic stability and cancer therapy.

In recent years, replication fork stability has emerged as an underlying mechanism of acquired drug resistance in *BRCA*-mutated cancers[3,18,19]. BRCA2-defective tumors become resistant to poly (ADP-ribose) polymerase (PARP) inhibitors or cisplatin by restoring fork stability without recovering homologous recombination (HR) mediated by BRCA2[3,20]. Thus, *BRCA*-mutated cancer cells rely on alternative fork stabilization mechanisms for survival upon drug-induced replication stress[18]. Therefore, increasing fork instability burden through targeted inhibition of compensatory fork stabilizing factors is a promising therapeutic approach to target human cancers with *BRCA2* loss-of-function mutations. Supporting this notion, pharmacological PARP inhibition exacerbates pathological fork degradation and cell death in BRCA2-deficient cells exposed to the replication inhibitor hydroxyurea (HU)[21].

WRN, a member of the RECQ family of DNA helicases[22], plays essential roles at stalled forks to counteract replication stress, thereby ensuring genomic stability[23–27]. However, owing to its dual helicase and exonuclease activities and multiple protein interactions, the functions of WRN at perturbed replication forks are complex and diverse. While WRN helicase activity promotes DNA2-mediated replication restart[26], the exonuclease function[28] or structural (noncatalytic) role of WRN[29] has been implicated in fork protection. How WRN operates at stalled forks upon deficiency of major fork stabilization mechanisms is largely unexplored.

In this study, we report that WRN helicase rescues stalled forks and limits hyper-degradation of nascent DNA in BRCA2-deficient cells. At the biochemical level, we show that WRN ATPase/helicase function promotes fork restoration and inhibits MRE11-mediated nucleolytic degradation of reversed forks. Accordingly, WRN helicase complementation rescues fork restart-delay in BRCA2-depleted cells. Moreover, WRN helicase or exonuclease partially rescues hyper-degradation of stalled forks caused by BRCA2 loss. Attenuation of WRN helicase results in enhanced fork degradation and defective fork restart in *BRCA2*-mutated cancer cells. Targeted WRN helicase inhibition causes rapid sequestration of WRN protein on chromatin and destabilizes stalled forks in BRCA2-deficient cells in a manner dependent on fork remodeling enzymes and the structure-specific nuclease MRE11. Consistent with these findings, *BRCA2*-mutated cancer cells exhibit increased DNA damage accumulation and elevated chromosomal instability upon WRN helicase inhibition. Pharmacological inhibition of WRN helicase preferentially kills BRCA2-deficient cancer cells and sensitizes them to the PARP inhibitor olaparib. In vivo, we show that WRN helicase inhibition retards *BRCA2*[−/−] tumor growth and induces DNA damage in a xenograft mouse model. Altogether, the results demonstrate that WRN helicase plays a specialized role in stabilizing forks experiencing stress that is deprotected due to BRCA2 deficiency.

## Results

**WRN stabilizes replication forks in BRCA2-deficient cells and ensures fork recovery.** Experimental evidence suggests that when a major fork protector such as BRCA2 is absent, other fork metabolizing proteins assume roles to stabilize stressed replication forks[8,17,30,31]. Given the apparent involvement but poorly understood function(s) of WRN during replication stress, we investigated its role in stalled fork stabilization when BRCA2 function is lost. Using the single-molecule DNA fiber assay, we first examined fork recovery following HU-induced replication stalling in *WRN* CRISPR knock-out (KO) (*WRN*[−/−]) U2OS cells upon siRNA-mediated knockdown of BRCA2 (Fig. 1a, b, and Supplementary Fig. 1a). As depicted in Fig. 1b, BRCA2-depleted U2OS/*WRN*[−/−] cells transfected with either control empty vector (EV) or a WRN expression construct (Fig. 1a) were subjected to a fork restart assay[15,26]. To evaluate fork restart efficiency upon release from HU-induced stalling, we determined the relative IdU/CldU ratio in the defined genetic conditions. Compared to control siRNA-treated cells (Median IdU/CldU ratio = 0.8376), BRCA2-depleted cells transfected with EV showed a significant decrease in IdU/CldU ratio (Median IdU/CldU ratio = 0.6279) (Fig. 1b), suggesting compromised replication recovery upon BRCA2 loss in *WRN*-deficient cells. To assess whether WRN can rescue the observed fork restart defect induced by loss of BRCA2, we examined fork recovery in BRCA2-depleted cells complemented with either wild-type (WT) or catalytic domain site-directed mutants of WRN (Fig. 1a, b, and Supplementary Fig. 1a). Exogeneous expression of WRN in BRCA2-depleted U2OS/*WRN*[−/−] cells fully restored the IdU/CldU ratio (Median IdU/CldU ratio = 0.8406) (Fig. 1b), suggesting efficient recovery of stalled forks ensured by WRN under the cellular condition of BRCA2 deficiency. However, the expression of helicase-dead WRN-K577M was unable to rescue the stalled fork phenotype evidenced by no significant change in median IdU/CldU ratio (0.6233) in BRCA2-depleted cells (Fig. 1b). In contrast, expression of WRN exonuclease-deficient mutant (WRN-E84A) rescued the fork restart defect (median IdU/CldU ratio = 0.7684). However, as compared to EV transfected cells, no significant change in median IdU/CldU ratio (0.6196) was observed in cells expressing a WRN mutant deficient in both helicase and exonuclease activities (WRN-K577M/E84A) (Fig. 1b). In agreement with these results, purified recombinant WRN-WT or WRN-E84A catalyzed fork restoration from a model reversed fork ("chicken-foot") DNA substrate in vitro whereas WRN-K577M failed to do so (Fig. 1c and d), suggesting fork restoration is supported by WRN ATPase/helicase activity. These results suggest that restoration of intact replications forks by WRN helicase is critical for efficient replication resumption following fork stalling when BRCA2 function is lost.

Defective replication restart is linked to uncontrolled nucleolytic processing of stalled forks if they are not sufficiently protected[13,15,32]. WRN has been previously shown to protect camptothecin (CPT)-induced stalled forks from MRE11/EXO1-mediated degradation in cells expressing WT BRCA2[28,29]. Unprotected reversed forks are extensively degraded by MRE11 exonuclease[33]. We reasoned that fork restoration by WRN helicase (Fig. 1d) may result in decreased reversed fork degradation by MRE11 nuclease because of the reduced availability of the reversed fork substrates. We, therefore tested recombinant MRE11 3′-5′exonuclease activity on a reversed fork substrate in the presence of recombinant WRN-E84A to monitor MRE11 digestion without interference caused by WRN

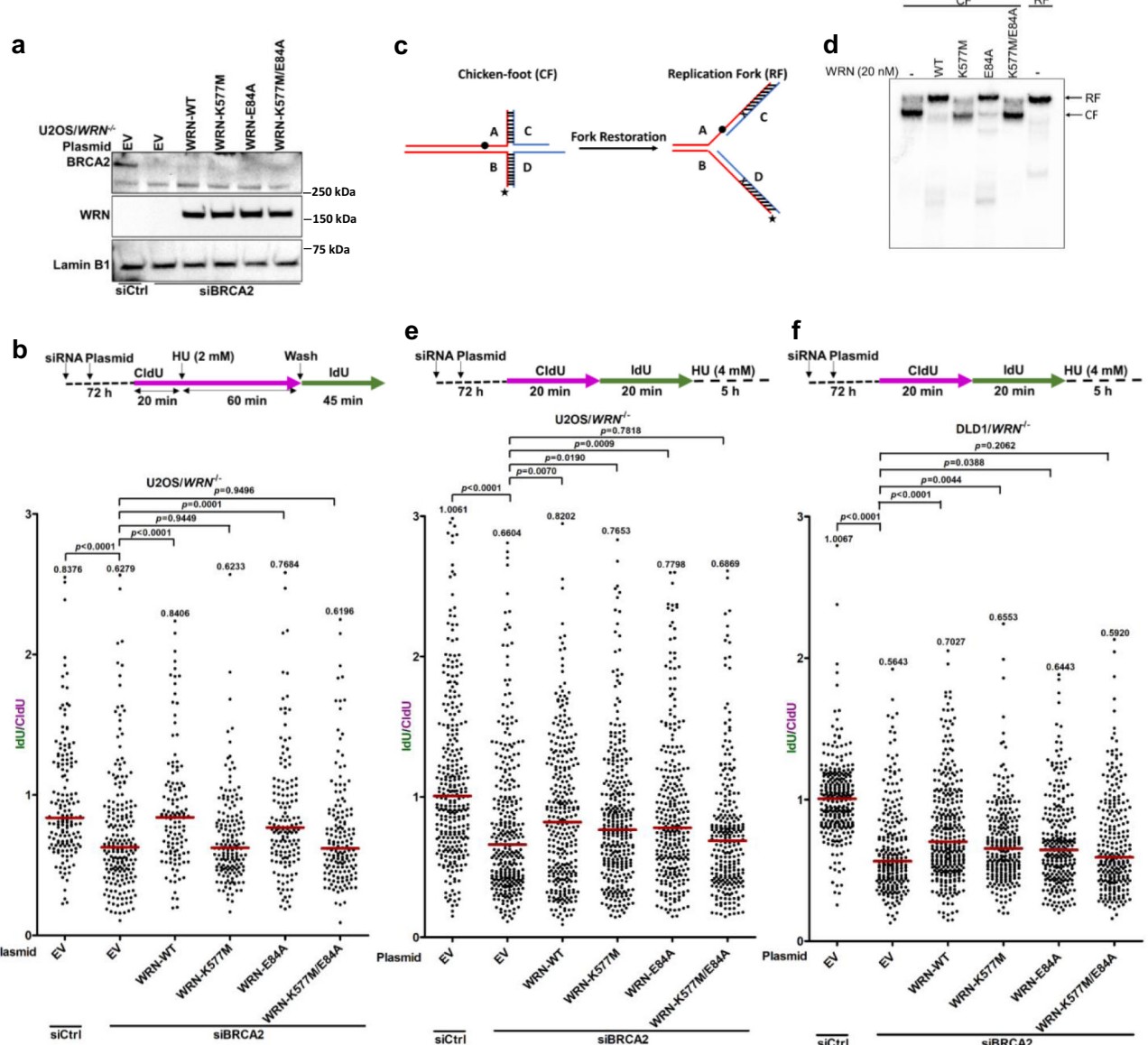

**Fig. 1 WRN helicase rescues fork restart defects and restrains hyper-degradation of stalled forks in BRCA2-deficient cells. a** Immunoblots showing BRCA2 and WRN protein levels in BRCA2-depleted U2OS/*WRN*$^{-/-}$ cells transfected with the indicated plasmids. **b** Fork restart assay in BRCA2-depleted U2OS/*WRN*$^{-/-}$ cells complemented with WT or catalytic dead WRN. Scatter plot showing IdU/CldU ratios in individual experimental conditions. Representative of $n = 2$ independent experiments; p-values ($p < 0.0001$, $p < 0.0001$, $p = 0.9449$, $p = 0.0001$, $p = 0.9496$) were derived from $n \geq 125$ DNA fibers using two-tailed Mann–Whitney test. **c** Schematic of the in vitro fork restoration assay showing "chicken-foot" (CF) and replication fork (RF) structures. Star indicates radiolabeled ($[\gamma\text{-}^{32}\text{P}]$ ATP) at 5′ DNA end. The black dot indicates the approximate position of an isocytosine base paired with a guanine on the opposite strand whereas the 2 X's in the RF indicate mismatched bases. Both modifications serve to minimize spontaneous conversion of the CF to the RF structure. **d** WRN ATPase/helicase activity catalyzes replication fork restoration in vitro. Fork restoration assay with 2 nM CF DNA substrate was performed using 20 nM of WT or catalytic mutant WRN proteins (lanes 2-5). Reactions using CF (lane 1) and RF (lane 6) substrates without WRN were used as negative and positive controls, respectively. Experiment was repeated three ($n = 3$) times with similar results. **e**, Quantification of fork stability assay performed in cells as described in a. Representative of $n = 3$ independent experiments; p-values ($p < 0.0001$, $p = 0.0070$, $p = 0.0190$, $p = 0.0009$, $p = 0.7818$) were derived from $n \geq 250$ DNA fibers using two-tailed Mann–Whitney test. **f**, Quantification of fork stability assay performed in BRCA2-depleted DLD1/*WRN*$^{-/-}$ cells upon complementation with WT or catalytic dead WRN. Representative of $n = 2$ independent experiments; p-values ($p < 0.0001$, $p < 0.0001$, $p = 0.0044$, $p = 0.0388$, $p = 0.2062$) were derived from $n \geq 230$ DNA fibers using two-tailed Mann–Whitney test. In **b**, **e**, and **f**, horizontal red bars indicate median of IdU/CldU ratios; median IdU/CldU values are indicated; purple and green colors indicate CldU and IdU labeling, respectively; data points within the specified axis limits are shown. Lamin B1 was used as loading control in immunoblots. Western blots (**a**) were repeated independently at least two times with similar results. Source data are provided as a Source Data file.

exonuclease on the substrate (Supplementary Fig. 1b). Digestion of the reversed fork DNA substrate by MRE11 was found to be substantially reduced in the presence of WRN-E84A, suggesting WRN helicase activity curtails MRE11-mediated fork degradation in vitro.

Next, we sought to determine if WRN could restrain uncontrolled nucleolytic degradation of unprotected forks in BRCA2-depleted cells. We, therefore, assessed the ability of WT or catalytic mutants of WRN to complement elevated fork instability induced by BRCA2 loss upon replication stalling by

HU (Fig. 1e and Supplementary Fig. 1c). For this, we performed fork stability assays in which BRCA2-depleted *WRN* KO cells exogenously expressing WT or WRN catalytic domain mutants were sequentially labeled with CldU and IdU nucleoside analogs followed by incubation with HU (4 mM) for 5 h (Fig. 1e and Supplementary Fig. 1c). With this experimental set-up, relative shortening of the IdU label compared to the corresponding CldU label of an ongoing (CldU–IdU) replication tract (IdU/CldU tract length ratio) was used to determine the level of nascent DNA degradation at stalled forks. Complementation with WRN-WT partially rescued the hyper-degradation phenotype caused by BRCA2 depletion in U2OS/*WRN*$^{-/-}$ cells (Fig. 1e). Expression of either WRN-K577M or WRN-E84A catalytic mutant could partially rescue fork degradation phenotype in BRCA2-depleted cells, whereas the WRN-K577M/E84A double mutant did not (Fig. 1e). We obtained similar results in DLD1/*WRN*$^{-/-}$ cells in which WRN-WT, WRN-K577M, or WRN-E84A, but not WRN-K577M/E84A, partially reduced the extensive fork degradation caused by BRCA2 silencing (Fig. 1f and Supplementary Fig. 1d). From biochemical assays with purified recombinant WRN protein and model DNA substrate, we found that both WRN-WT and WRN-K577M/E84A double mutant could bind to synthetic reversed fork DNA substrate with similar affinity (Supplementary Fig. 1e), suggesting loss of both catalytic activities rather than loss of DNA binding prevented the WRN double mutant from rescuing hyper-degradation phenotype due to BRCA2 deficiency. Of note, we observed partial fork degradation upon complementation with either WRN-WT or WRN-E84A in control siRNA-transfected U2OS/*WRN*$^{-/-}$ cells (Supplementary Fig. 2a), suggesting limited nucleolytic processing of the stalled forks by WRN helicase when BRCA2 is present, as reported previously[26]. Collectively, these results suggest that WRN limits excessive fork degradation upon BRCA2 loss wherein both catalytic functions of WRN are involved.

BRCA2 regulates nucleolytic processing of stalled forks[3,4,34–36]. In BRCA2-deficient cells, extensive fork degradation is mostly carried out by MRE11 or EXO1 nucleases[35], whereas DNA2 nuclease and WRN helicase are involved in limited reversed fork resection leading to fork restart in BRCA2-proficient cells[26]. However, restoration of WRN function in BRCA2-depleted cells did not further upregulate fork resection. Rather WRN partially rescued nascent DNA tract length (Fig. 1e and f). Therefore, the rescue of fork restart defects in BRCA2-depleted cells upon WRN complementation (Fig. 1b) is consistent with the role of WRN helicase in the restoration of the active replication fork from the reversed fork state (Fig. 1d). As defects in replication fork restart may lead to reduced cell survival[15], we examined the clonogenic potential of isogenic WT or *WRN*$^{-/-}$ U2OS cells depleted of BRCA2. Clonogenic survival assays demonstrated a significantly compromised survival of BRCA2-depleted *WRN*$^{-/-}$ cells (Supplementary Fig. 2b), suggesting that combined loss of WRN and BRCA2 is toxic to human cells. Of note, as judged by Western blot analysis of total cell lysate, BRCA2 level was reduced in *WRN*$^{-/-}$ U2OS cells (Supplementary Fig. 2b) and DLD1 cells (Supplementary Fig. 2c), suggesting that phenotypes associated with WRN deficiency may reflect in part a loss of BRCA2; however, this requires further study.

**WRN helicase inhibition exacerbates replication fork instability in *BRCA2*-mutated cancer cells**. Having demonstrated that WRN complementation partially rescued the hyper-degradation phenotype of BRCA2-depleted cells (Fig. 1e and f), we tested the effect of WRN knockdown on nascent DNA degradation in the *BRCA2*-mutated ovarian cancer cell line PEO1[37]. Consistent with the well-documented role of BRCA2 in

fork protection[3,4], treatment of PEO1 cells with 4 mM HU for 3 h resulted in reduced fork stability (Median IdU/CldU = 0.6773) compared to WT *BRCA2* revertant PEO4 cells[37] (Median IdU/CldU = 0.8369) (Fig. 2a and Supplementary Fig. 2d). However, nascent strand degradation was even further elevated (Median IdU/CldU = 0.6126) by WRN depletion in the *BRCA2*-mutated cells (Fig. 2a). In contrast, WRN knockdown in WT cells significantly rescued nascent DNA tract length (Median IdU/CldU = 0.9176), consistent with the involvement of WRN in resection of stalled forks in BRCA2-proficient cells as described previously[26].

We further examined fork stability in *BRCA2*-mutated cells exposed to a small molecule inhibitor of WRN helicase (WRNi) NSC617145[38] (Fig. 2b and Supplementary Fig. 2e). Treatment with HU (4 mM for 3 h) alone resulted in a significant ($p < 0.0001$) level of fork degradation in PEO1 cells (Median IdU/CldU = 0.6833) as compared to isogenic PEO4 cells with restored BRCA2 function[37] (Median IdU/CldU = 0.9166) (Fig. 2b). However, the stability of the unprotected forks in *BRCA2*-mutated cells was severely compromised upon treatment with WRNi (4 μM), as we observed a significant decrease in median IdU/CldU ratio (0.3831) (Fig. 2b). RECQL1 overexpression did not affect fork instability induced by the WRNi in PEO1 cells (Supplementary Fig. 3a), suggesting the in vivo abundance of RECQL1 does not alter the fork-destabilizing effects of NSC617145. Elevated nascent DNA degradation was also observed in *BRCA2*$^{-/-}$ DLD1 cells pre-treated with WRNi (Supplementary Fig. 3b). Notably, the exacerbated fork instability in *BRCA2*-mutated cells upon acute inhibition of WRN helicase by NSC617145 (Median IdU/CldU = 0.3831) was found to be markedly greater (Fig. 2b) than genetic depletion of *WRN* altogether (Median IdU/CldU = 0.6126) (Fig. 2a), suggesting that WRN helicase inhibition triggers unrestrained degradation of stalled forks in BRCA2-deficient human cancer cells. Because DNA2 nuclease operates in conjunction with WRN helicase to allow limited fork processing in BRCA2 WT cells[26], we tested the stability of HU-stalled forks upon DNA2 inhibition in *BRCA2*-mutated cells using a previously characterized small molecule inhibitor of DNA2 nuclease[39] (Supplementary Fig. 3c). Unlike WRN helicase inhibition, DNA2 inhibition did not exacerbate fork degradation in *BRCA2*-mutated PEO1 cells; moreover, the DNA2i did not synergize with the WRNi in its effect on fork stability (Supplementary Fig. 3c). WRN helicase inhibition was previously reported to cause mitomycin C hypersensitivity for cells lacking FANCD2[38], a protein important for fork stability and implicated in the Fanconi anemia pathway of interstrand crosslink repair[8]. Therefore, we tested whether WRN helicase inhibition triggers elevated fork instability in FANCD2-deficient cells. Unlike BRCA2-deficient cells, WRN helicase inhibition did not significantly exacerbate fork degradation in *FANCD2*$^{-/-}$ PD20 cells (Supplementary Fig. 3d), suggesting that the role of WRN helicase activity in fork maintenance is distinct when BRCA2 or FANCD2 is absent.

**Loss of WRN or inhibition of its helicase activity compromises replication restart in *BRCA2*-mutated cancer cells**. Next, we examined the efficiency of stalled fork recovery in WT or *BRCA2*-mutated cells either depleted of WRN (Fig. 2c and Supplementary Fig. 4a) or exposed to the WRNi (Fig. 2d and Supplementary Fig. 4b). Consistent with previously published results[3,35], *BRCA2*-mutated PEO1 cells exhibited shorter restarted tract length (Median IdU/CldU = 1.2121) compared to the WT PEO4 cells (Median IdU/CldU = 1.4607) (Fig. 2c). In WT cells, knockdown of WRN reduced IdU/CldU tract length ratio (Median IdU/CldU = 1.1083), suggesting a role of WRN in fork recovery as

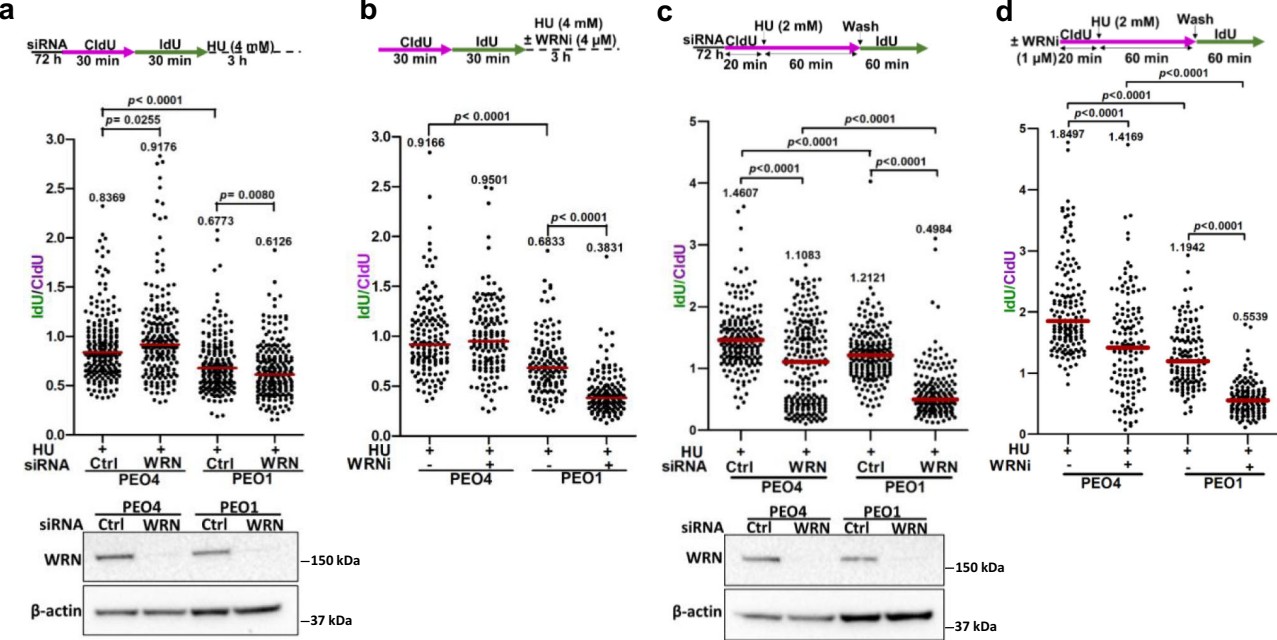

**Fig. 2 WRN helicase inhibition causes increased fork degradation and replication restart defects in *BRCA2*-mutated cancer cells. a** (Upper panel) Schematic of the fork stability assay performed in PEO1 and PEO4 cells upon WRN knockdown. (Lower panel) Scatter dot plot showing quantification of IdU/CldU ratios in individual experimental conditions. Representative of $n = 3$ independent experiments; $p$-values ($p = 0.0255$, $p < 0.0001$, $p = 0.0080$) were derived from $n \geq 200$ DNA fibers using two-tailed Mann–Whitney test. Knockdown of WRN was verified by immunoblotting. **b** (Upper panel) Schematic of the fork degradation experiment in PEO1 and PEO4 cells upon WRNi treatment. (Lower panel) Graphical quantification of the results obtained from the fork stability experiments as described above. Representative of $n = 2$ independent experiments; $p$-values ($p < 0.0001$, $p < 0.0001$) were derived from $n \geq 125$ DNA fibers using two-tailed Mann–Whitney test. **c** (Upper panel) Scheme of the fork restart assay performed in PEO1 and PEO4 cells upon siRNA-mediated knockdown of WRN. (Lower panel) Scatter plot showing IdU/CldU ratios in individual experimental conditions. Representative of $n = 2$ independent experiments; $p$-values ($p < 0.0001$, $p < 0.0001$, $p < 0.0001$, $p < 0.0001$) were derived from $n \geq 195$ DNA fibers using two-tailed Mann–Whitney test. Immunoblots showing relative knockdown levels of WRN as assessed 72 h post-transfection. **d**, (Upper panel) Schematic of the fork restart assay in HU-treated PEO1 and PEO4 cells upon WRN helicase inhibition. (Lower panel) Scatter plot showing IdU/CldU tract length ratios in individual experimental conditions. Representative of $n = 2$ independent experiments; $p$-values ($p < 0.0001$, $p < 0.0001$, $p < 0.0001$, $p < 0.0001$) were derived from $n \geq 125$ DNA fibers using two-tailed Mann–Whitney test. Horizontal red bars in scatter plots indicate the median of IdU/CldU ratios; median IdU/CldU values are indicated; purple and green colors indicate CldU and IdU labeling, respectively; data points within the specified axis limits are shown. β-actin was used as a loading control in immunoblots in (**a**) and (**c**). Western blots **a**, **c** were repeated independently at least two times with similar results. Source data are provided as a Source Data file.

described previously[25,26]. Moreover, WRN depletion in *BRCA2*-mutated cells showed an even more pronounced reduction in IdU/CldU ratio (Median IdU/CldU = 0.4984) compared to the isogenic WRN-depleted WT cells, indicating severely compromised fork recovery under the condition of WRN and BRCA2 co-deficiency.

Compared to WT PEO4 cells (Median IdU/CldU = 1.4169), *BRCA2*-mutated PEO1 cells exposed to WRNi exhibited a nearly three-fold reduction in IdU/CldU ratio (Median IdU/CldU = 0.5539) (Fig. 2d), suggesting dramatically inefficient or delayed fork restart under conditions of BRCA2 deficiency and pharmacological inhibition of WRN helicase. To test the specificity of the WRNi, we performed fork restart assays after NSC617145 exposure using $WRN^{-/-}$ cells (Supplementary Fig. 4c) or $RECQL1^{-/-}$ cells (Supplementary Fig. 4d). While WRN deficiency markedly attenuated the defect in WRNi-induced fork restart (Supplementary Fig. 4c), we observed a significant reduction in fork restart efficiency in $RECQL1^{-/-}$ cells upon WRNi treatment (Supplementary Fig. 4d), indicating specificity of the WRNi to modulate fork restart. Consistent with a published role of RECQL1 in fork restart[40], we observed significantly reduced fork restart efficiency in $RECQL1^{-/-}$ cells compared to isogenic $RECQL1^{-/-}$ cells corrected by exogenous expression of RECQL1-WT (Supplementary Fig. 4d). Exposure of

$RECQL1^{-/-}$ cells to NSC617145 exasperated the fork restart defect (Supplementary Fig. 4d), suggesting a specific effect of the WRNi that is independent of RECQL1 status.

**WRNi triggers MRE11-mediated nascent DNA degradation in *BRCA2*-mutated cancer cells in a manner dependent on cellular fork reversal DNA translocases.** Given the enhanced degradation of HU-stalled forks in *BRCA2*-mutated cells exposed to the WRNi (Fig. 2b), we tested if NSC617145 alone could trigger nascent DNA degradation (Fig. 3a). We observed significant ($p < 0.0001$) reduction of the IdU/CldU ratio upon NSC617145 treatment in *BRCA2*-mutated cells (Median IdU/CldU = 0.6451) compared to WT cells (Median IdU/CldU ratio = 0.9381) (Fig. 3a), suggesting nascent DNA degradation. However, nascent strand shortening in WRNi-treated *BRCA2*-mutated cells was significantly ($p < 0.0001$) alleviated (Median IdU/CldU = 0.7977) upon siRNA-mediated knockdown of WRN (Fig. 3a), suggesting that NSC617145-induced nascent strand degradation phenotype is WRN-dependent.

In the absence of functional BRCA2, stalled forks are progressively degraded by MRE11 nuclease[4]. Therefore, we sought to determine whether the observed nascent DNA degradation upon NSC617145 treatment is mediated by

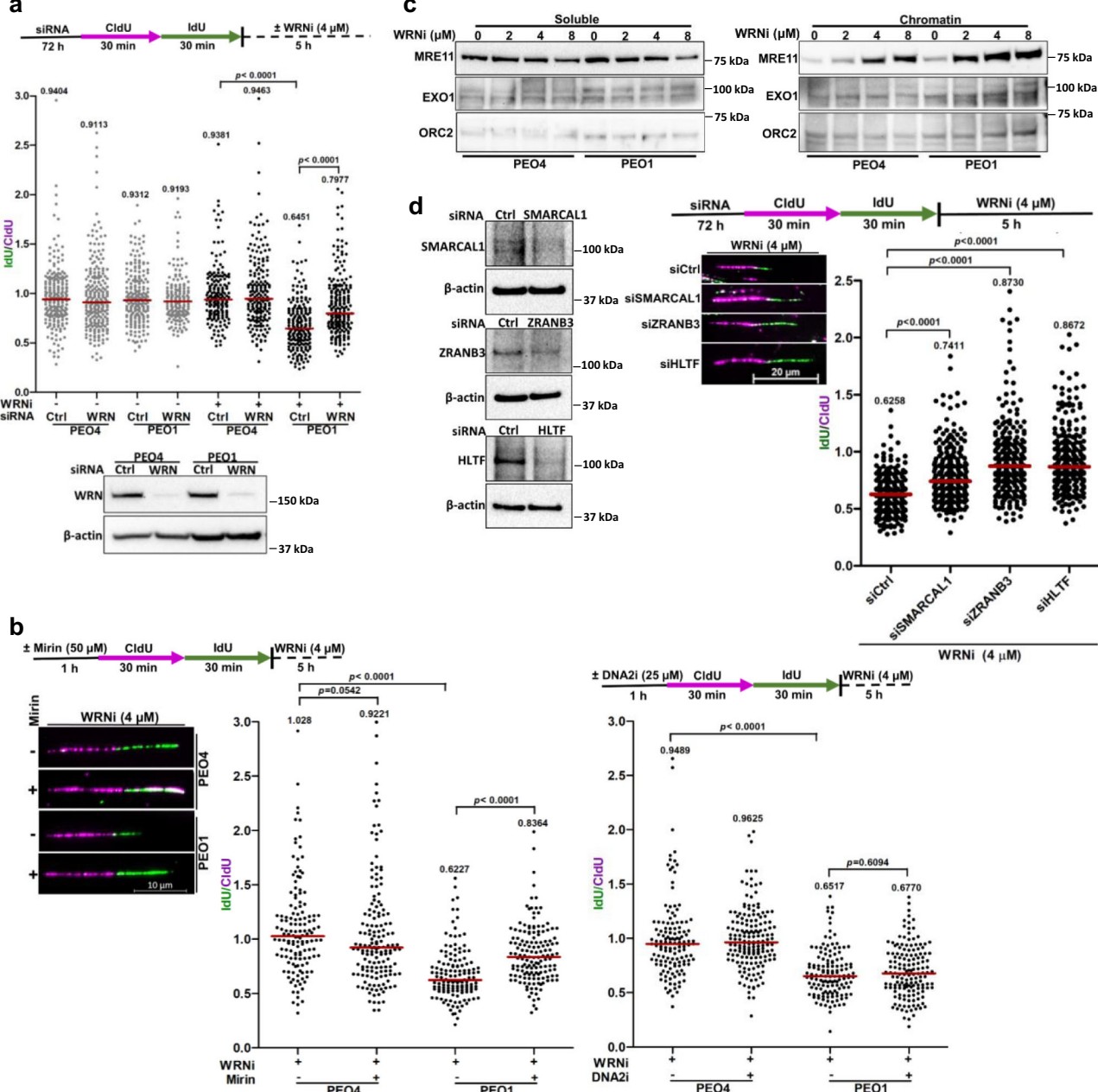

**Fig. 3 WRN helicase inhibition triggers SNF2 translocase-dependent nascent DNA degradation by MRE11 nuclease in *BRCA2*-mutated cancer cells.**
**a** Fork stability assay in PEO1 and PEO4 cells exposed to WRNi NSC617145. Cells transfected with either control or WRN siRNA (80 nM) were sequentially labeled with CldU and IdU and subjected to WRNi (NSC617145) treatment (4 μM) for 5 h. For untreated control, cells were collected immediately after labeling. Scatter plot showing IdU/CldU tract length ratios in individual experimental conditions. Representative of *n* = 2 independent experiments; *p*-values (*p* < 0.0001, *p* < 0.0001) were derived from *n* ≥ 150 DNA fibers using two-tailed Mann–Whitney test. Immunoblots showing WRN knockdown. **b** Fork stability assay in NSC617145-treated PEO1 and PEO4 cells upon Mirin (left panel) or DNA2i (right panel) treatment. Representative fibers of the Mirin experiment are shown. Representative of *n* = 2 independent experiments; *p*-values (*p* = 0.0542, *p* < 0.0001, *p* < 0.0001, *p* < 0.0001, *p* = 0.6094) were derived from *n* ≥ 125 DNA fibers using two-tailed Mann–Whitney test. **c** Immunoblots showing dose-dependent chromatin enrichment of MRE11 upon WRNi treatment. Cells were treated with indicated doses of WRNi (NSC617145) for 1 h and subjected to subcellular fractionation. EXO1 level is shown. ORC2 was used as a positive marker and loading control for chromatin fractions. **d** Fork stability assay in PEO1 cells depleted of SMARCAL1, ZRANB3, or HLTF upon NSC617145 treatment. (Left panel) Immunoblots showing knockdown. Representative DNA fibers are shown. (Right panel) Quantification of IdU/CldU ratios in individual experimental conditions. Representative of *n* = 2 independent experiments; *p*-values (*p* < 0.0001, *p* < 0.0001, *p* < 0.0001) were derived from *n* ≥ 250 DNA fibers using two-tailed Mann–Whitney test. In **a**, **b**, and **d** horizontal red bars indicate the median of IdU/CldU ratios; median IdU/CldU values are indicated; purple and green colors indicate CldU and IdU labeling, respectively. β-actin was used as a loading control in immunoblots in (**a**) and (**d**). Western blots **a**, **c**, **d** were repeated independently at least two times with similar results. Source data are provided as a Source Data file.

MRE11. Fork stability in *BRCA2*-mutated PEO1 cells exposed to NSC617145 was substantially rescued upon treatment with the MRE11 nuclease inhibitor Mirin[41], with a statistically significant ($p < 0.0001$) increase in median IdU/CldU ratio (Fig. 3b). IdU/CldU ratio was not significantly altered in WT PEO4 cells upon MRE11 nuclease inhibition (Fig. 3b). Unlike the case for MRE11, pre-exposure to a small molecule inhibitor of DNA2 nuclease[39] did not rescue WRNi-induced fork degradation in PEO1 cells (Fig. 3b), suggesting that MRE11 plays a more dominant role in nucleolytic degradation of stalled forks in WRNi-treated *BRCA2*-mutated cells. Consistent with this notion, MRE11 was preferentially enriched in the chromatin fraction obtained from PEO1 cells exposed to WRNi compared to PEO4 cells (Fig. 3c). In BRCA2-deficient cells, degradation of HU-stalled fork initiated by MRE11 is further extended by EXO1 nuclease[35]. However, we did not observe chromatin enrichment of EXO1 following WRNi treatment (Fig. 3c). MRE11 and EXO1 may independently act on stalled forks in BRCA-deficient cells, as combined loss of the two nucleases has been reported to further rescue fork stability in these cells[35]. Our results suggest that WRNi causes rapid accumulation of replication fork intermediates suitable for MRE11 engagement and subsequent degradation of nascent DNA if not protected by the fork-stabilizing protein BRCA2.

In response to replication stress, stalled forks frequently undergo reversal to form four-way chicken foot-like structures[42]. SNF2-like DNA translocases (SMARCAL1, ZRANB3, and HLTF) have been implicated in driving fork reversal under conditions of replication stress in human cells[20,42,43]. As reversed forks are vulnerable to nuclease-mediated degradation upon BRCA loss, attenuation of fork reversal has been shown to restore fork stability in BRCA-deficient cells[20,33]. We sought to determine whether observed fork degradation in *BRCA2*-mutated cells occurs downstream of SNF2 translocase-dependent fork reversal upon WRNi treatment. Therefore, we determined relative fork stability upon siRNA-mediated knockdown of SMARCAL1, ZRANB3, and HLTF in *BRCA2*-mutated PEO1 cells exposed to 4 μM NSC617145. PEO1 cells depleted of SMARCAL1, ZRANB3 or HLTF exhibited a significant ($p < 0.0001$) rescue in median IdU/CldU tract length ratio, indicating restoration of fork stability (Fig. 3d). These results suggest that WRNi-induced nascent DNA degradation by MRE11 in *BRCA2*-mutated cancer cells is preceded by DNA translocase-mediated fork remodeling.

**WRN helicase inhibition causes fork stalling and asymmetry in BRCA2-deficient cells.** We next assessed the effect of WRN helicase inhibition on replication fork progression. Following 15 min of CldU pulse labeling to mark actively replicating genomic regions, U2OS cells were subsequently labeled with IdU for another 60 min (Supplementary Fig. 5a). WRNi (2 μM) was added concomitantly with IdU labeling. Quantification of the data obtained from fiber spread analysis revealed a significant ($p < 0.0001$) reduction in IdU/CldU ratio following NSC617145 exposure, suggesting a delay in fork progression. Thus, NSC617145 alone impedes replication fork progression in the absence of genotoxic stress. However, the compound did not affect fork progression in U2OS $WRN^{-/-}$ cells (Supplementary Fig. 5a), suggesting that the replication defects triggered by NSC617145 are WRN-dependent.

To assess the effect of WRN helicase inhibition on fork progression as a function of BRCA2 status, we examined fork progression rate in WT and $BRCA2^{-/-}$ DLD1 cells exposed to NSC617145. IdU tract lengths were significantly reduced ($p < 0.0001$) in both genotypic backgrounds by NSC617145 treatment (Fig. 4a and Supplementary Fig. 5b). Replication fork asymmetry was quantified by measuring relative lengths of sister

forks (SFs) emanating from individual bi-directional origins (Fig. 4b). Compared to untreated cells, WRNi-treated cells exhibited increased SF asymmetry demonstrated by a significant reduction in SF ratios for WT cells ($p = 0.0046$) and $BRCA2^{-/-}$ cells ($p < 0.0001$), respectively (Fig. 4b), suggesting frequent replication stalling or collapse. Importantly, fork asymmetry was further increased ($p = 0.0035$) in $BRCA2^{-/-}$ cells as compared to WT cells exposed to WRNi (Fig. 4b). Compared to WT cells, we observed a relatively greater percentage of asymmetric forks (SF ratio < 0.6) in $BRCA2^{-/-}$ cells (Fig. 4c). To substantiate these results, we monitored replication fork dynamics in isogenic WT PEO4 and *BRCA2*-mutated PEO1 cells upon WRNi treatment. In addition to a significant reduction in fork progression rate (Supplementary Fig. 5c), we observed increased fork asymmetry in WRNi-treated *BRCA2*-mutated PEO1 cells (Supplementary Fig. 5d, e), corroborating the results obtained with DLD1 cells.

**WRNi NSC617145 directly binds WRN and traps it on chromatin.** To further characterize the biological effect of NSC617145, we tested for a direct interaction of the $^{14}$C-radiolabeled compound (Supplementary Fig. 6a) to WRN in vitro and in human cells. $^{14}$C-NSC617145 was incubated with purified recombinant WRN (Supplementary Fig. 6b), and the mixture was applied to a nitrocellulose filter. Detection of radiolabel bound to the nitrocellulose demonstrated that there was approximately a 9.5-fold greater $^{14}$C signal in binding mixtures containing WRN compared to no WRN (Fig. 5a). Binding mixtures containing the sequence-related RECQL1 helicase (Fig. 5a) or the Superfamily 2 helicase FANCJ (Supplementary Fig. 6c) displayed a low $^{14}$C signal comparable to that of the binding mixture lacking WRN, suggesting specific binding of $^{14}$C-NSC617145 to WRN in vitro. To assess the binding of NSC617145 to WRN in vivo, U2OS $WRN^{-/-}$ cells transiently expressing FLAG-tagged WRN were incubated with $^{14}$C-NSC617145 followed by immunoprecipitation using anti-FLAG antibody (Fig. 5b). A 3-fold greater $^{14}$C signal in immunoprecipitates prepared from WRN-expressing cells was detected compared to those from cells expressing only the FLAG epitope (Fig. 5b), indicating $^{14}$C-NSC617145 binding to WRN expressed in human cells.

Because targeted WRN helicase inhibition triggered greater fork degradation than that induced by WRN depletion (Fig. 2a and b), we hypothesized that cellular exposure to NSC617145 generates static WRN–DNA complexes that interfere with DNA replication. To address this, we assessed the amount of chromatin-bound WRN in cells exposed to NSC617145 and observed its dose-dependent enrichment (Fig. 5c). Compared to WT cells, NSC617145-induced chromatin enrichment of WRN was elevated in *BRCA2*-mutated cells, suggesting that WRN accumulates at forks when BRCA2 is absent (Fig. 5d). Furthermore, even in the absence of NSC617145, WRN showed enrichment in the chromatin fraction of *BRCA2*-mutated cells (Fig. 5d). To test if this was related to fork stalling or general DNA damage processing, we examined WRN immunofluorescence intensity in PEO1 and PEO4 replicating cells counterstained with the S-phase-specific marker PCNA (Supplementary Fig. 6d). As reported previously[44], we observed that WRN primarily localized to nucleoli in the unperturbed condition. However, compared to WT cells, PCNA-positive *BRCA2*-mutated cells showed increased WRN intensity in nucleoplasmic foci, suggesting WRN accumulation at forks in the absence of BRCA2. Of note, consistent with published data that actively proliferating cancer cell lines display an elevated level of WRN[45], we detected substantially enhanced WRN immunofluorescence in PCNA-positive cells as compared to PCNA-negative cells

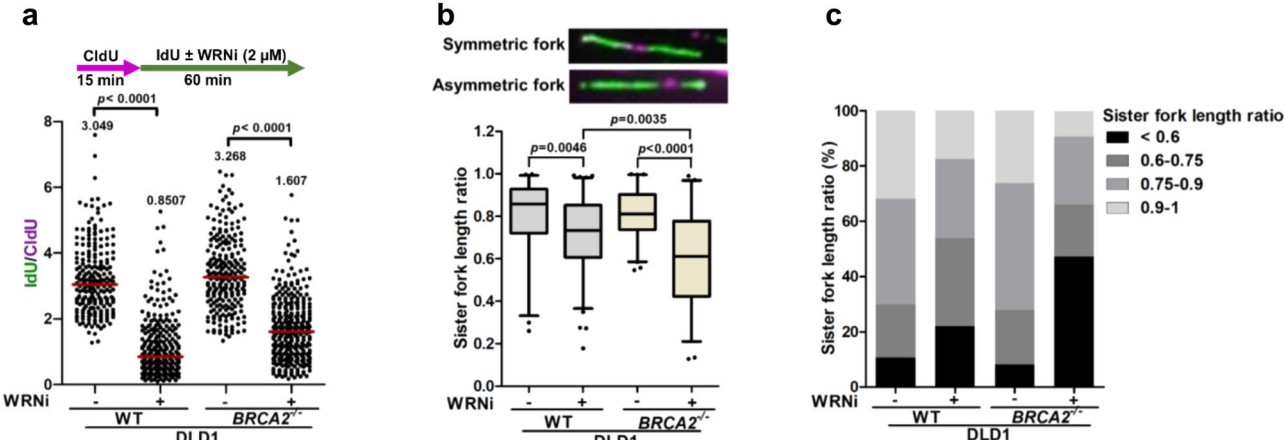

**Fig. 4 WRN helicase inhibitor causes replication fork stalling. a** (Upper panel) Schematic of the fork progression assay in WT and *BRCA2*−/− DLD1 cells upon WRNi treatment. (Lower panel) Scatter dot plot showing IdU/CldU ratios in individual genotypes treated or untreated with the WRNi. Representative of *n* = 3 independent experiments; *p*-values (*p* < 0.0001, *p* < 0.0001) were derived from *n* ≥ 200 DNA fibers using two-tailed Mann–Whitney test. **b** (Upper panel) Representative DNA fibers depicting symmetric and asymmetric bi-directional replication forks. Length of the two sister forks (IdU) emanating from a single bidirectional origin (CldU) was measured from individual bidirectional (green-purple-green) fork structures scored from the experiment as described in **a**. (Lower panel) Boxplot (5–95th percentile) of sister fork (SF) ratios (short to long IdU tract length ratio) in WT and *BRCA2*−/− DLD1 cells treated or untreated with WRNi. Bounds of the box represent 25th–75th percentile, middle horizontal lines show the median, whiskers indicate 5th–95th percentiles. Representative of *n* = 3 independent experiments; *p*-values (*p* = 0.0046, *p* = 0.0035, *p* < 0.0001) were derived from *n* ≥ 40 bidirectional forks using two-tailed Mann–Whitney test. **c** Bar graph showing percent distribution of sister fork length ratios. In **a**, horizontal red bars indicate the median of IdU/CldU ratios; median IdU/CldU values are indicated; purple and green colors indicate CldU and IdU labeling, respectively. Source data are provided as a Source Data File.

(Supplementary Fig. 6d). NSC617145-induced chromatin binding of WRN was further enhanced in the presence of HU (Fig. 5e), suggesting greater accumulation of WRN-DNA complexes when replication forks stall due to nucleotide depletion in vivo. To assess whether WRN trapping by NSC617145 is reversible, we examined the relative amount of chromatin-bound WRN upon drug removal (Fig. 5f). Cellular exposure to NSC617145 led to a substantial increase in chromatin-bound WRN which remained even 2.5 h after drug removal, suggesting persistent trapping of WRN (Fig. 5f). Importantly, neither DNA topoisomerase I (Fig. 5f) nor DNA2 nuclease (Supplementary Fig. 6e) was found to accumulate in the chromatin fraction in NSC617145-treated cells, suggesting a specific effect of the compound to bind and trap WRN on genomic DNA.

**WRN helicase inhibition causes MUS81-dependent DSB formation, activates nonhomologous end-joining (NHEJ), and elevates genome instability in *BRCA2*-mutated cancer cells.** Defective processing of stalled forks under conditions of replication stress results in DNA damage accumulation and increased chromosomal instability in BRCA1/2-defective cells[3,8,20]. Therefore, we examined if WRN helicase inhibition induced DSBs in *BRCA2*-mutated cancer cells. *BRCA2*-mutated PEO1 cells exhibited a significantly (*p* = 0.0275) greater percentage of γH2AX positive nuclei (Mean % γH2AX positive nuclei = 31.4) than WT PEO4 cells (Mean % γH2AX positive nuclei = 20.7) upon NSC617145 exposure (Supplementary Fig. 7a). We obtained similar results in DLD1 cells in which there was a 2-fold greater increase in Mean % γH2AX foci positive nuclei in *BRCA2*−/− cells (44.3%) compared to WT nuclei (18.7%) (Supplementary Fig. 7b). These observations were validated by Western blotting which showed increased γH2AX in WRNi-treated *BRCA2*−/− cells compared to WT cells (Supplementary Fig. 7c). To substantiate these results, we examined the focal accumulation of 53BP1, a marker of DSBs, in cells exposed to WRNi (Supplementary Fig. 7d). Percent 53BP1 positive nuclei were found to be

approximately 3-fold greater (*p* = 0.0045) in *BRCA2*−/− cells treated with WRNi (48.4%) compared to WT cells (16.6%) exposed to the WRNi (Supplementary Fig. 7d), indicating enriched DNA damage accumulation in *BRCA2*-deficient cells upon WRN helicase inhibition. To determine if DNA damage induced by exposure to the WRNi occurs preferentially in replicating cells, we co-stained for γH2AX and PCNA (Fig. 6a). γH2AX was predominantly observed in PCNA-positive nuclei, suggesting replication-associated DNA damage (Fig. 6a). PEO1 cells exposed to WRNi exhibited γH2AX foci in >80% of the PCNA-positive nuclei (Fig. 6a), suggesting increased DNA damage in *BRCA2*-mutated cells following WRN helicase inhibition.

Previously, it was shown that WRN loss of function results in replication fork collapse and subsequent formation of MUS81-mediated DSBs at collapsed forks[23]. Moreover, MUS81-dependent DSBs are generated following hyper-degradation of stalled forks in BRCA2-deficient cells[35]. Therefore, we assessed DSB formation upon WRN helicase inhibition by neutral Comet assay (Fig. 6b). Compared to WT PEO4 cells, *BRCA2*-mutated PEO1 cells displayed elevated DSBs upon WRNi treatment (Fig. 6b). However, increased DSB formation in *BRCA2*-mutated cells was found to be significantly abrogated upon MUS81 depletion (Fig. 6b and Supplementary Fig. 7e). These observations suggest that WRN helicase inhibition results in MUS81-dependent DSBs as a consequence of stalled fork hyper-degradation in *BRCA2*-mutated cells.

Two major DNA repair pathways, HR and NHEJ, compete for the processing of replication-associated DNA breaks. Cells deficient of major HR factors such as BRCA1/2 shift DNA repair towards the error-prone NHEJ pathway[46–48]. Accordingly, we anticipated that DNA damage accumulated in *BRCA2*-mutated cells upon WRN helicase inhibition might trigger aberrant activation of NHEJ. To address this hypothesis, we examined for NHEJ activation by DNA-PKcsThr2609 foci formation[49,50] in PEO1 and PEO4 cells treated with NSC617145. WT cells showed a small (~1.7-fold) but statistically significant (*p* = 0.0112) increase in the percentage of cells having DNA-PKcsThr2609

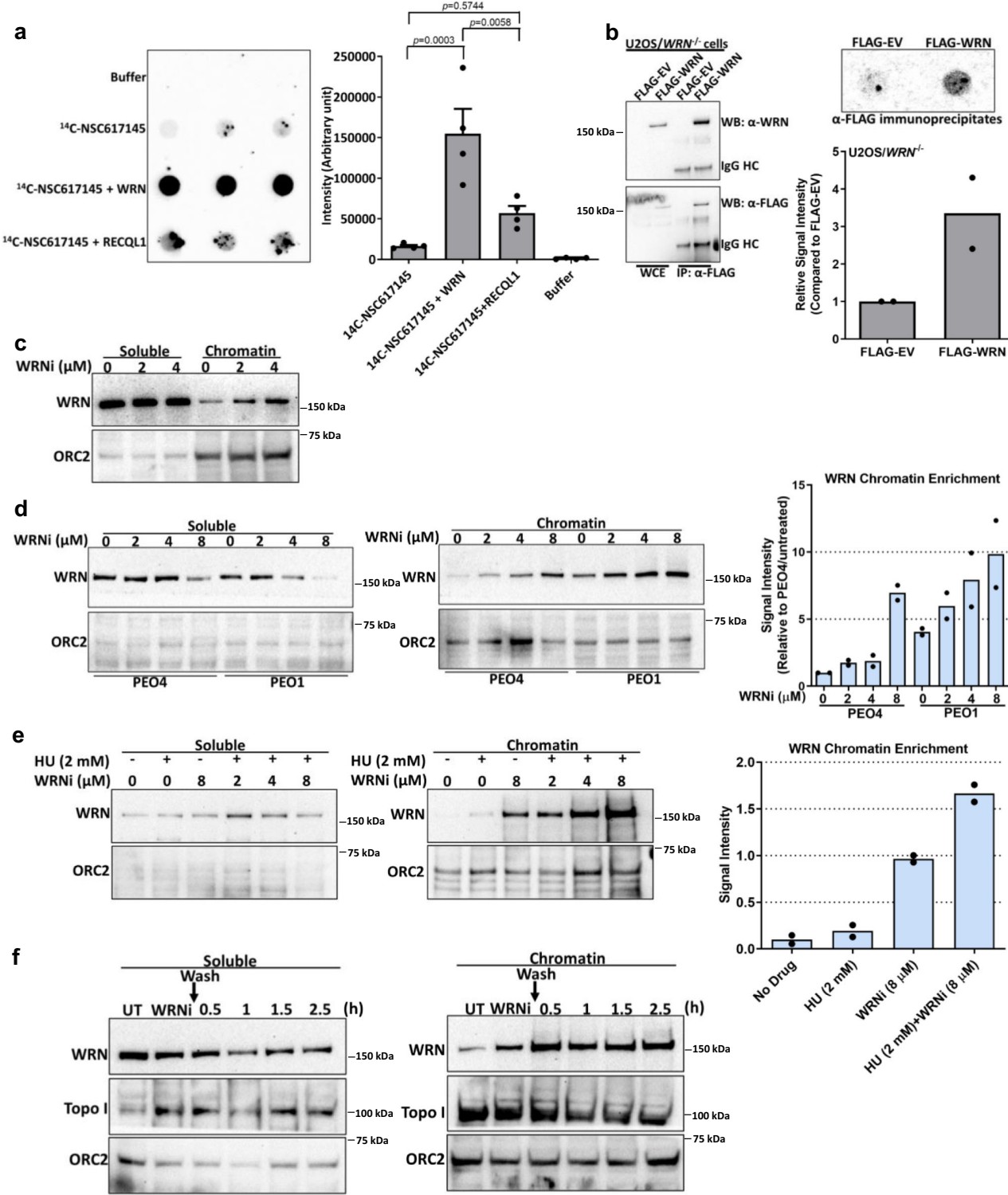

positive nuclei upon treatment with WRNi (Supplementary Fig. 8a). In contrast, we observed a >4-fold increase ($p = 0.0040$) in the percentage of DNA-PKcsThr2609 positive nuclei in *BRCA2*-mutated PEO1 cells upon WRN helicase inhibition, suggesting increased activation of the NHEJ pathway (Supplementary Fig. 8a). To determine if the aberrant NHEJ activation occurs in replicating cells, we monitored DNA-PKcs phosphorylation at serine 2056 (DNA-PKcs pS2056)[51] in cells co-stained for PCNA (Fig. 7a). We observed DNA-PKcs pS2056 formation primarily in PCNA-positive nuclei, suggesting WRNi-

induced NHEJ activation in cells actively performing DNA synthesis. Interestingly, DNA-PKcs pS2056 foci were reduced in *BRCA2*-mutated cells exposed to Mirin (Fig. 7a), suggesting induction of NHEJ is related to MRE11-mediated fork degradation. In BRCA2-deficient cells, hyper-degraded forks are rescued by break-induced replication (BIR) repair mechanism following MUS81-mediated cleavage[35]. Rad52, a key mediator of HR has been implicated in the repair of collapsed replication forks[52] and recovery of stalled forks[16]. We, therefore, analyzed the percentage of stalled forks in WRNi-treated PEO1 cells upon knockdown of

**Fig. 5 NSC617145 directly binds WRN and causes WRN trapping on DNA. a** Dot blots showing in vitro binding of radiolabeled $^{14}$C-NSC617145 (10 μM) to recombinant WRN or RECQL1 proteins (300 nM). Quantitation of the data obtained from the binding experiments. Data represent mean ± SEM of four repeats. One-way ANOVA with Bonferroni's multiple comparison test: $p = 0.0003$, $p = 0.0058$, $p = 0.5744$. **b** $^{14}$C-NSC617145 binds to WRN proteins in U2OS cells. Immunoblots showing WRN protein levels in whole-cell extract and α-FLAG immunoprecipitates prepared from U2OS/$WRN^{-/-}$ cells transfected with either empty FLAG vector or FLAG-WRN expression plasmids. IgG heavy chains (HC) are shown. α-FLAG immunoprecipitates prepared from $^{14}$C-NSC617145 (10 μM) treated cells were subjected to dot blot assay. The bar graph represents $^{14}$C-NSC617145 signal intensity. Data represent the mean of two independent experiments. **c** Immunoblots showing WRN chromatin enrichment in PEO1 cells treated with indicated doses of NSC617145 for 3 h. ORC2 was used as a positive marker and loading control for chromatin fractions. The experiment was repeated three times with similar outcomes. **d** Relative chromatin enrichment of WRN in PEO4 and PEO1 cells upon WRNi treatment for 1 h. Bar graph showing WRN levels in chromatin-bound fractions prepared from PEO4 and PEO1 cells. WRN signal intensity was normalized to ORC2 and represented as relative fold change over PEO4 untreated cells (set as one). Data represent the mean of two independent experiments. **e** Immunoblots showing enhanced chromatin enrichment of WRN upon NSC617145 exposure in presence of HU. PEO1 cells were treated with indicated doses of NSC617145 with or without HU (2 mM) for 2 h and subjected to sub-cellular fractionation. Bar graph represents quantification of WRN chromatin enrichment. Data represent the mean of two independent experiments. **f** Persistent trapping of WRN on chromatin by WRNi. PEO1 cells were treated with NSC617145 (6 μM) for 2 h before removal of the drug by ice-cold PBS wash. Cells were incubated further in a pre-warmed cell culture medium for the indicated times. Soluble and chromatin fractions were probed with the indicated antibodies. The experiment was repeated independently two times with similar results. Source data are provided as a Source Data file.

RAD52 (Supplementary Fig. 8b). Either WRN helicase inhibition or RAD52 depletion resulted in significantly elevated fork stalling. RAD52 knockdown in WRNi-treated PEO1 cells further increased the percentage of stalled forks, suggesting a role of RAD52-dependent BIR to aid in the recovery of hyper-degraded forks. However, given the observed increase in NHEJ (Supplementary Fig. 8a and Fig. 7a), our results collectively suggest that in *BRCA2*-mutated cells, a subset of collapsed forks downstream of WRNi-induced hyper-degradation also engage NHEJ for repair.

Elevated NHEJ leads to chromosomal aberrations under conditions of HR deficiency[46,48,53]. Moreover, experimental evidence suggests that uncontrolled degradation and subsequently delayed restart of stressed replication forks results in chromosomal instability in HR-deficient cells[4,8,15]. We, therefore tested whether inhibition of WRN helicase triggers genomic instability in *BRCA2*-mutated cells (Fig. 7b). Analysis of metaphase chromosome spreads prepared from WRNi-treated WT PEO4 cells did not reveal any significant change ($p = 0.3853$) in average chromosomal aberrations (Fig. 7b). However, structural aberrations including chromatid breaks, gaps or end-to-end fusions between heterologous chromosomes were elevated by ~1.7-fold ($p < 0.0001$) in *BRCA2*-mutated PEO1 cells exposed to the WRNi (Fig. 7b). Inhibition of MRE11 nuclease alleviated gross chromosomal aberrations in WRNi-treated *BRCA2*-mutated PEO1 cells (Fig. 7c), suggesting that WRNi-induced genomic instability was primarily attributed to MRE11-mediated fork degradation. We also observed a significant reduction in chromosomal abnormalities upon DNA-PK inhibition (Fig. 7c), suggesting NHEJ induction contributes to the heightened genomic instability in WRNi-treated *BRCA2*-mutated cells. Collectively, these observations suggest that blocking WRN helicase function pharmacologically elicits DNA damage and error-prone NHEJ, leading to genomic instability in cells lacking functional BRCA2.

**WRN helicase inhibition kills *BRCA2*-defective cancer cells.** Uncontrolled fork degradation and defective restart of stressed replication forks result in elevated genome instability leading to cell death[15,18,26,33,35]. Therefore, we tested the effects of WRN helicase inhibition on the survival of cancer cells lacking functional BRCA2. *BRCA2*$^{-/-}$ DLD1 cells displayed increased sensitivity to the WRNi as determined by colony survival and cell viability assays (Fig. 8a, Supplementary Fig. 9a-b). Selective killing of *BRCA2*$^{-/-}$ cells by PARP1 inhibitor (PARPi) olaparib treatment confirmed the BRCAness phenotype of these cell lines. We also observed a marked decrease in colony survival capacity of

BRCA2-depleted U2OS cells exposed to increasing concentrations of NSC617145 (Supplementary Fig. 9c). As a positive control, BRCA2-depleted U2OS cells were found to be selectively killed by olaparib, consistent with the well-established synthetic lethal interaction of PARP1 and BRCA2[54]. BRCA2-selective hypersensitivity of cancer cells towards the WRNi was further recapitulated in the breast cancer cell line MDA-MB-231 (Supplementary Fig. 10a) and ovarian cancer cell line SKOV-3 (Supplementary Fig. 10b), suggesting generality of the phenotype. At a 0.5 μM dose of NSC617145, BRCA2-depleted MDA-MB-231 and SKOV-3 cells exhibited >4-fold and >3-fold decrease in colony survival, respectively (Supplementary Fig. 10a, b). Although acute inhibition of WRN helicase by NSC61745 was found to be synergistic with *BRCA2* loss, we did not observe any significant effect on colony-forming ability of DLD1 *BRCA2*$^{-/-}$ cells upon siRNA-mediated knockdown of WRN (Supplementary Fig. 10c). However, no *WRN* CRISPR KO clones of *BRCA2*$^{-/-}$ DLD1 or *BRCA2*-mutated PEO1 cells were recovered, suggesting a synthetic lethal interaction of WRN and BRCA2. Presumably, DLD1 *BRCA2*$^{-/-}$ cells depleted of WRN by RNA interference remain viable due to residual WRN.

We next assessed if clinically relevant *BRCA2* pathogenic mutations rendered cancerous cells susceptible to WRN helicase inhibition. For this, we compared the relative viability of *BRCA2*-mutated ovarian cancer cell line PEO1 with that of *BRCA2* WT revertant PEO4 cells upon WRN helicase inhibition. Compared to WT PEO4 cells, we observed drastically reduced survival of *BRCA2*-mutated PEO1 cells exposed to WRNi (Fig. 8b and Supplementary Fig. 11a, b). At 0.75 μM WRNi, hardly any PEO1 colonies were detected whereas ~70% of PEO4 colonies remained viable (Fig. 8b). Notably, *BRCA2*-mutated PEO1 cells were not as sensitive to olaparib as *BRCA2*$^{-/-}$ DLD1 (Fig. 8a and Supplementary Fig. 9a, b), as evident from colony survival (Fig. 8b) and cell viability assay data (Supplementary Fig. 11b). These observations are consistent with previous findings that some *BRCA2*-mutated ovarian cancer cell lines such as PEO1 are partially resistant to PARPi[55].

RAD51 functions downstream of BRCA2 in the homology-directed repair of DNA DSBs as well as in the fork protection pathway[56]. We, therefore anticipated that like BRCA2, RAD51 depletion could have similar effects on cellular fitness upon WRN helicase inhibition. Indeed, knockdown of RAD51 sensitized DLD1 cells to WRNi treatment as revealed by results obtained from colony survival (Supplementary Fig. 12a) and cell viability (Supplementary Fig. 12b) assays. We were curious if NSC617145-induced cell killing was dependent on WRN. To address this, we examined the survival of WRN-depleted cells after NSC61745

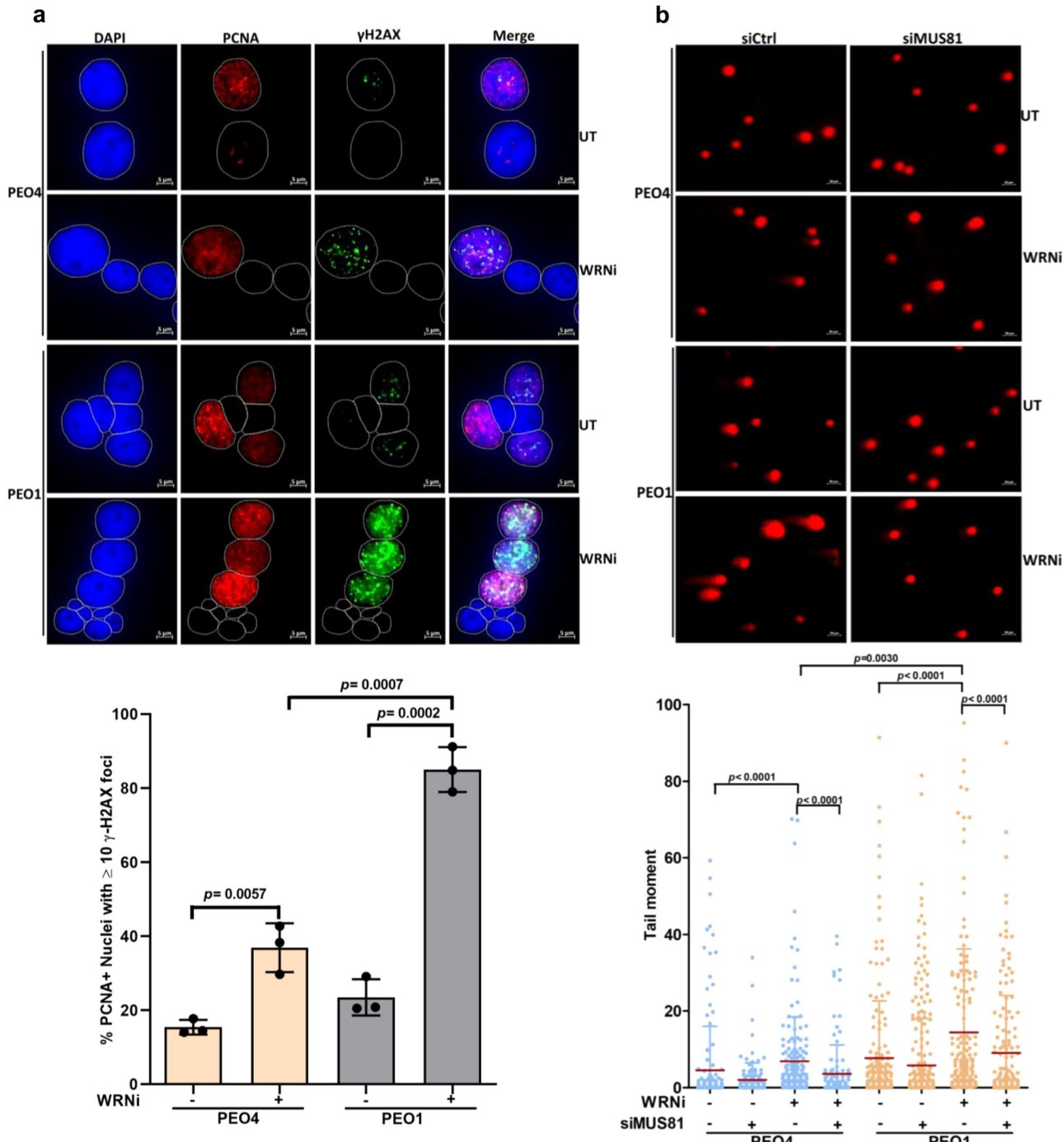

**Fig. 6 WRNi triggers MUS81-dependent double-strand break formation in *BRCA2*-mutated cancer cells. a** Immunofluorescence staining of γH2AX following treatment of PEO4 and PEO1 cells with WRNi (5 μM) for 6 h. Cells were co-stained with PCNA to identify replicating cells. Nuclei were stained with DAPI. DAPI, PCNA, and γH2AX are shown in blue, red, and green, respectively. Scale bars are shown. Bar graph represents quantification of % PCNA-positive cells with ≥10 γH2AX foci under untreated (UT) and WRNi-treated conditions. Data represent mean ± SD of three independent experiments; two-tailed unpaired *t*-test; *p* values are indicated. **b** Analysis of DSB formation using neutral Comet assay. Control or MUS81-depleted PEO4 and PEO1 cells were subjected to neutral comet assay following treatment with WRNi (5 μM) for 6 h. Representative images of comets obtained from individual experimental conditions are shown. Scale bars are shown. Scatter dot plot showing tail moment in individual samples. Data represent mean ± SD of n ≥ 120 comets analyzed under each condition over n = 2 independent experiments. Mean tail moments (horizontal red bars) ± SD are shown. ≥120 comets were analyzed for each condition using the OpenComet plugin in ImageJ. Two-tailed Mann–Whitney test; *p* values (*p* < 0.0001, *p* < 0.0001, *p* = 0.0030, *p* < 0.0001, *p* < 0.0001) are indicated. Source data are provided as a Source Data file.

exposure. Colony survival in response to NSC617145 was significantly rescued in WRN-depleted HeLa and DLD1 cells (Supplementary Fig. 13a, b), suggesting WRN-dependent cytotoxicity of the inhibitor. Substantially improved survival of

$WRN^{-/-}$ U2OS cells exposed to NSC617145 (Supplementary Fig. 13c, d) further attested to the specificity of the compound. However, isogenic *RECQL1*-deficient and *RECQL1*-corrected cells exhibited similar sensitivity to the WRNi (Supplementary

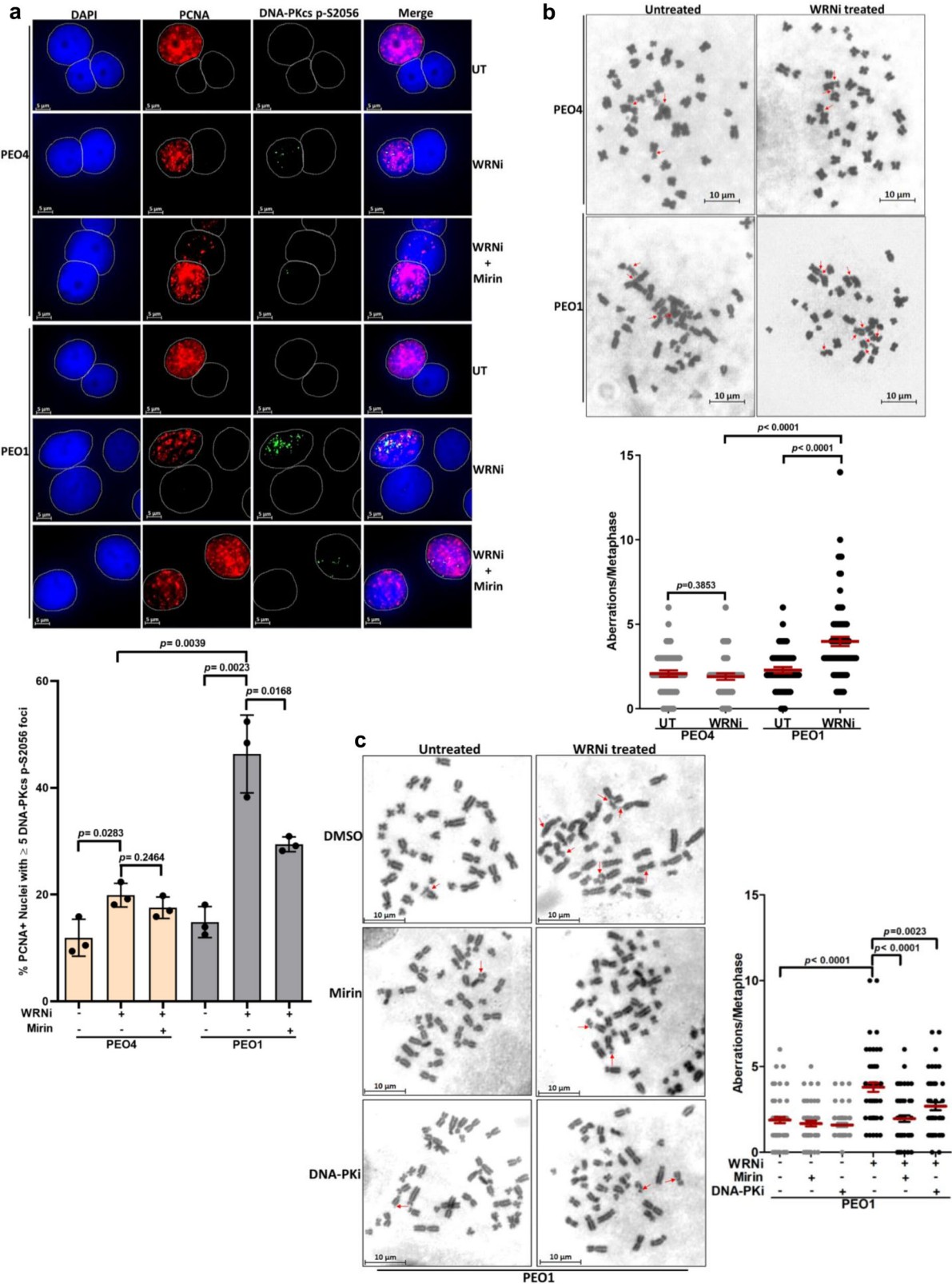

Fig. 13e), suggesting a RECQL1-independent effect. Hypersensitivity to WRNi upon BRCA2 depletion was also found to be attenuated in $WRN^{-/-}$ cells (Supplementary Fig. 13f). These observations are consistent with our previously published results demonstrating WRN dependency of NSC617145 for its effect on cell survival[38].

**Effect of NSC617145 on *BRCA2*-deficient tumor growth and DNA damage in vivo.** Given the observed hypersensitivity of a broad range of BRCA2-defective cancer cells to NSC617145, we tested the in vivo response of *BRCA2*-deficient tumors to the WRNi using xenograft tumor models in athymic nude mice (Fig. 8c). Mice carrying DLD1 WT or $BRCA2^{-/-}$ tumors were

**Fig. 7 Elevated nonhomologous end-joining and genomic instability in *BRCA2*-mutated cancer cells exposed to WRNi. a** Representative immunofluorescence staining of DNA-PKcs p-S2056 foci in PEO4 and PEO1 cells following treatment with WRNi (5 μM) for 16 h. Cells were optionally treated with Mirin (25 μM). Cells were co-stained with PCNA to mark replicating cells. Bar graph represents quantification of % PCNA positive cells with ≥5 DNA-PKcs p-S2056 foci in individual experimental conditions. Data represent mean ± SD of three independent experiments; two-tailed unpaired *t*-test; *p* values are indicated. **b** Representative images of metaphase chromosome spread prepared from PEO1 and PEO4 cells treated or non-treated with WRNi (2 μM) for 48 h. Scatter plot showing relative chromosomal aberrations scored from the cells as described above. Data represent mean ± SEM of *n* ≥ 40 metaphase spreads analyzed under each condition over *n* = 3 independent experiments; Two-tailed Mann–Whitney test; *p* values ($p = 0.3853$, $p < 0.0001$, $p < 0.0001$) are indicated. **c** Analysis of chromosomal aberrations in *BRCA2*-mutated PEO1 cells following treatment with WRNi (5 μM) for 16 h. Cells were optionally treated with Mirin (25 μM) or DNA-PK inhibitor (DNA-PKi) NU7441 (10 μM). Quantification of chromosomal aberration analysis is shown. Data represent mean ± SEM of *n* ≥ 50 metaphase spreads analyzed under each condition over *n* = 3 independent experiments; Two-tailed Mann–Whitney test; *p* values ($p < 0.0001$, $p < 0.000$, $p = 0.0023$) are indicated. Structural aberrations including chromatid breaks/gaps, chromosome end-to-end fusions, dicentric and radial chromosomes are indicated by red arrows in **b** and **c**. Scale bars are shown in all representative images. In **a**, DAPI, PCNA, and DNA-PKcs p-S2056 foci are shown in blue, red, and green, respectively. Source data are provided as a Source Data file.

treated with either NSC617145 or drug vehicle alone and relative tumor growth was monitored over time. Compared to control vehicle treatment, treatment with NSC617145 (25 mg/kg body weight) resulted in a marked decrease ($p = 0.05$) in *BRCA2*$^{-/-}$ tumor growth (Fig. 8c). Importantly, NSC617145 treatment exhibited >3-fold higher growth inhibitory effect on *BRCA2*$^{-/-}$ tumor (TGI = 34.70%) as compared to WT tumors (TGI = 10.28%). Thus, the in vivo data are consistent with results from cell-based assays demonstrating preferential killing of BRCA2-deficient cancer cells by WRN helicase inhibition. Furthermore, NSC617145 treatment led to elevated DNA damage accumulation in *BRCA2*$^{-/-}$ tumor cells as demonstrated by increased immunohistochemical staining for γH2AX (Fig. 8d) and 53BP1 (Fig. 8e) in tissue sections of *BRCA2*$^{-/-}$ tumors compared to WT tumors.

**Pharmacological WRN helicase inhibition sensitizes *BRCA2*-mutated cancer cells to olaparib.** *BRCA1/2*-mutated tumors often exhibit de novo or acquired PARPi resistance[57]. Particularly, ovarian cancers with recurrent *BRCA*-mutations show incomplete responses to clinical PARP inhibitors[58]. Consistent with the previously published results[55], we also observed that *BRCA2*-mutated ovarian cancer cells PEO1 showed reduced sensitivity to the PARPi olaparib (Fig. 8b and Supplementary Fig. 11b). We anticipated that treatment with sublethal doses of WRNi could potentiate olaparib cytotoxicity in *BRCA2*-mutated cancer cells. Olaparib treatment was not found to be very effective in killing PEO1 cells as ~60% colonies were still viable even at the highest concentration of drug tested (0.5 μM) (Fig. 9a and Supplementary Fig. 14a). In contrast, co-treatment with 0.25 μM NSC617145 led to ~86% reduction in PEO1 colony survival at a 10-fold lower olaparib concentration (0.05 μM) (Fig. 9a). Although the apparently synergistic cytotoxic effect of the WRNi and olaparib was also observed in WT PEO4 cells, it was much less pronounced as compared to PEO1 cells and ~40% of the WT PEO4 colonies remained viable at even the highest dose of olaparib (0.5 μM) (Fig. 9a). We further assessed drug–drug interaction effects by analyzing the coefficient of drug interaction (CDI) and observed synergistic cell killing (CDI < 1) by the WRNi-olaparib combination in both PEO1 and PEO4 cells (Fig. 9b). However, at the lowest concentration of olaparib (0.05 μM), there was a much stronger synergism (CDI < 0.7) in PEO1 cells compared to PEO4 cells (Fig. 9b). Bliss synergy analysis further revealed synergistic cell killing by WRNi and olaparib as indicated by positive Excess over Bliss (EOB > 0) synergy scores (Fig. 9c). *BRCA2*$^{-/-}$ colorectal cancer cells DLD1 co-treated with olaparib and NSC617145 also displayed a greater reduction in proliferation than that due to an additive effect of the two drugs acting independently (Supplementary Fig. 14b). However, NSC617145 treatment did not sensitize WT DLD1 cells

to olaparib, suggesting BRCA2-selective synergistic cytotoxicity of WRNi and olaparib. These results demonstrate that pharmacological inhibition of WRN helicase potentiates PARPi cytotoxicity in BRCA2-defective cancer cells.

## Discussion

Replication stalling, estimated to occur at 3,000 of 12,000 active forks per nucleus in human cells[59,60], is a serious threat to genomic stability if not dealt with in a timely fashion[61]. In response to mild genotoxic stress, it is estimated that 15–30% of stalled forks are remodeled to reversed fork structures to allow template repair and prevent DSBs[60,62]. Stabilization of stalled replication forks is a complex process involving multiple fork remodelers and protectors. Although WRN is demonstrably active in stalled fork processing, its mechanism of action at stalled and deprotected forks has remained enigmatic. Here, we identified a crucial role of WRN helicase to rescue stalled replication forks in cancer cells under genetic conditions of *BRCA2* deficiency. WRN is required to enact a timely restart following replication stalling and limit excessive degradation of stalled forks in *BRCA2*-deficient cancer cells. WRN ATPase/helicase efficiently restored a model replication fork from reversed fork state in vitro and rescued fork restart defects caused by BRCA2 loss. DNA fiber experiments further demonstrated that loss or inhibition of WRN helicase activity resulted in severely compromised fork restart in *BRCA2*-mutated cells, suggesting WRN helicase, but not exonuclease, is required for timely fork restart in *BRCA2*-deficient cells. In contrast, both helicase and exonuclease functions of WRN partially rescued nascent strand degradation under the BRCA2-depleted condition. Unlike *BRCA2*-proficient cells, WRN depletion in *BRCA2*-deficient cells did not rescue nascent DNA tract length, but instead further enhanced nascent strand degradation upon replication stalling, suggesting WRN limits hyperdegradation of unprotected forks. In agreement with these results, we observed reduced nucleolytic cleavage of a synthetic reversed fork substrate by MRE11 nuclease in the presence of WRN helicase in vitro. Thus, our results support a model in which stalled fork restoration by WRN helicase limits hyperdegradation of reversed forks by reducing substrate availability for MRE11 degradation in BRCA2-deficient cells (Fig. 9d). Rad52-dependent BIR promotes stalled fork recovery under conditions of replication stress[16]. Moreover, in *BRCA2*-deficient cells, resected reversed forks are rescued by a POLD3-dependent BIR mechanism following MUS81-mediated fork cleavage[35]. Thus, although not formally tested, it is plausible that WRN and RAD52-MUS81-POLD3 are two axes used by cells to restart forks in a *BRCA2*-deficient background.

We determined that WRN exonuclease partially rescued the hyper-degradation of unprotected forks in BRCA2-depleted cells. Previously, it was reported that in BRCA2-proficient cells WRN

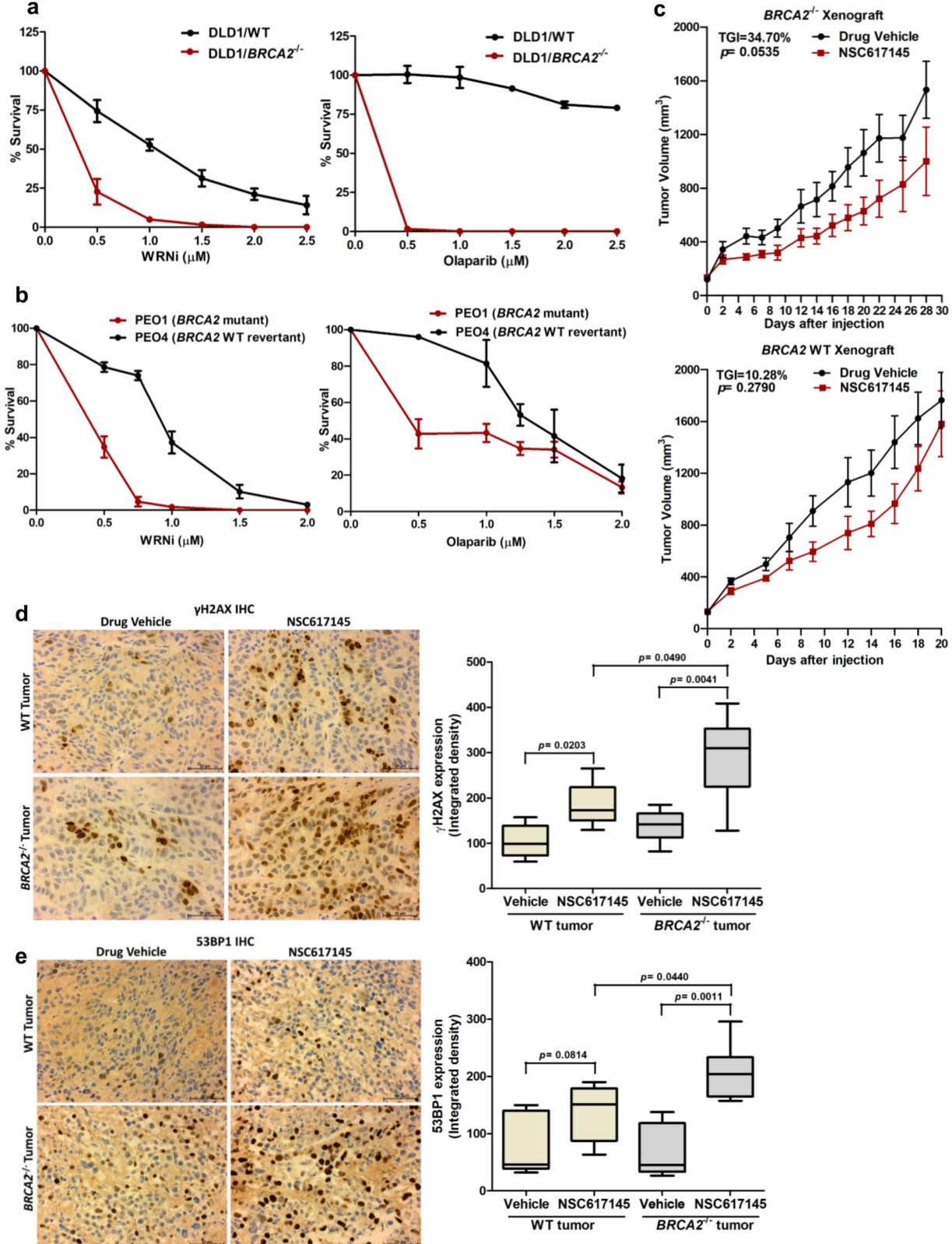

exonuclease activity suppresses nascent strand DNA degradation upon fork stalling induced by CPT[28]. The importance of WRN exonuclease in fork protection was attributed to the limitation of extensive nucleolytic activity by MRE11 and EXO1, thereby suppressing chromosomal instability[28]. Thus, WRN exonuclease limits extensive fork degradation in both the WT and genetically deficient BRCA2 cells. Interestingly, CtIP which is an auxiliary

factor of MRE11 in HR repair also protects forks during replication stress via its nuclease activity[11].

The current study provides molecular insight into the role of WRN at deprotected forks. BRCA2 ensures fork protection by loading Rad51 onto exposed single-stranded DNA at the site of the stalled fork. However, our findings suggest that WRN also plays a role in controlling nucleolytic processing when stalled fork

**Fig. 8 WRNi kills BRCA2-deficient cancer cells and retards *BRCA2*$^{-/-}$ xenograft tumor growth in mice. a** Colony survival of WT and *BRCA2*$^{-/-}$ DLD1 cells exposed to WRNi (left panel) or olaparib (right panel). WT and *BRCA2*$^{-/-}$ DLD1 cells were plated on a six-well plate at 1,500/well and exposed to the indicated doses of WRNi or olaparib 4 h after cell seeding. **b** Line graphs showing relative colony survival of PEO1 and PEO4 cells exposed to increasing doses of WRNi (left panel) or olaparib (right panel). PEO1 and PEO4 cells were plated on a six-well plate at 1,000 cells/well and treated with the indicated doses of WRNi or olaparib 4 h after cell seeding. In **a** and **b**, cells were grown in presence of WRNi or olaparib for 12 days before colony staining. Data represent mean ± SD of three independent experiments. **c** Line graph showing a relative growth rate of *BRCA2*$^{-/-}$ (upper panel) and WT (lower panel) xenograft tumors in athymic nude mice treated with either drug vehicle or NSC61745. After the tumors reached the palpable size, mice were injected intraperitoneally with either drug vehicle or NSC617145 (25 mg/kg body weight) every alternate day for a total of 14 days. Mean tumor volume ± SD ($n = 6$) at each time point is shown. Tumor growth inhibition (TGI) is indicated. *p*-values were determined using two-tailed unpaired Welch's *t*-test. **d** and **e** Immunohistochemical (IHC) staining for γH2AX (**d**) and 53BP1 (**e**) expression in tissue sections obtained from vehicle or NSC617145 treated wild type and *BRCA2*$^{-/-}$ xenograft tumors. Representative images (×40 magnification) of DAB-stained tissue sections are shown. Scale bar represents 50 μM. Box-whisker plots showing the median integrated density of DAB staining analyzed from $n = 5$ tumor tissue samples under each condition. Bounds of the box represent 25th–75th percentile, middle horizontal lines show the median, whiskers indicate maximum and minimum values of the dataset; *p* values were calculated using two-tailed unpaired *t*-test. Middle horizontal lines of the boxes represent the median. Source data are provided as a Source Data file.

structures (e.g., reversed fork) are deprotected. Furthermore, experimental results in our study demonstrate that inhibition of the structure-specific nuclease MRE11 helps to curtail excessive nucleolytic degradation of nascently synthesized DNA caused by WRN helicase inhibition in *BRCA2*-mutated cells. Controlling nucleolytic degradation at the stalled fork is paramount to fork restart so that DNA replication can be completed to its entirety in a timely manner. Collectively, our results suggest an important function of WRN catalytic functions (helicase and exonuclease) to prevail when there is a genetic deficiency of *BRCA2*.

Targeted inhibition of WRN helicase triggers heightened fork instability and reduces survival of *BRCA2*-mutated human cancer cells. Furthermore, we found that pharmacological inhibition of WRN helicase destabilizes replication forks to a greater extent than WRN depletion in vivo, suggesting an additional burden imposed by NSC617145-induced WRN helicase inhibition compared to loss of WRN altogether. Using biochemical and cell-based assays, we found that NSC617145 binds WRN and exerts its biological effects in a WRN-dependent manner, confirming its target specificity. Intriguingly, the selective and persistent sequestration of WRN to chromatin in cells exposed to WRNi suggests that WRN becomes trapped on genomic DNA. Therefore, the increased fork instability induced by NSC617145 is likely the combined effects of WRN helicase inhibition and static WRN–DNA complexes that interfere with fork progression. There are important classes of anticancer drugs including Topoisomerase inhibitors and PARP inhibitors that confer cytotoxicity by both catalytic inhibition and trapping of their target enzymes[54]. However, this is an instance of a small molecule inhibitor targeted against a human DNA helicase that operates by a trapping mechanism to interfere with replication fork dynamics. It is plausible that other protein partners are also sequestered with the trapped WRN–DNA complex; however, there was not a significant enrichment of DNA2 or Topoisomerase I, consistent with the specificity of the WRNi. Replication fork slowing induced by the WRNi was alleviated upon WRN depletion, suggesting static WRN–DNA complexes impeded fork progression as well.

We further demonstrated that WRNi-induced stalled forks undergo MRE11-mediated nucleolytic degradation in *BRCA2*-mutated cancer cells. Nascent strand degradation occurs downstream of reversed fork formation catalyzed by the SNF2 fork remodeling enzymes SMARCAL1, ZRANB3, and HLTF[20,33]. Therefore, restoration of fork stability upon knockdown of SNF2 translocases in WRNi-treated *BRCA2*-mutated cells suggests that pharmacological inhibition of WRN helicase leads to nascent DNA degradation preceded by translocase-dependent fork remodeling (Fig. 9e). Consistent with this idea, WRN helicase inhibition resulted in a strong chromatin enrichment of MRE11

that contributes to fork degradation. WRN loss results in fork collapse and MUS81-dependent DSBs following replication stalling[23]. Also, in *BRCA2*-deficient cells, partially resected forks are cleaved by MUS81 endonuclease leading to transient accumulation of DSBs[35]. In agreement with this, we found that WRNi treatment contributes to DSBs in *BRCA2*-mutated cells in a MUS81-dependent manner, suggesting WRNi-induced stalled forks undergo MUS81 cleavage following nascent DNA degradation. Our results demonstrate that fork stalling represents a rapid and global response to WRN helicase inhibition in human cells and that the consequential stalled forks require BRCA2 for stabilization. Uncontrolled fork degradation destabilizes stalled forks leading to defective fork restart which contributes to irreversible fork collapse, genomic instability, and cell death[4,18,20,63]. Thus, our results suggest that in *BRCA2*-deficient cells, forks stalled by static WRN–DNA complexes collapse leading to elevated chromosomal instability and cell death (Fig. 9e).

Fork stability has been linked to acquired chemoresistance in *BRCA2*-mutated human tumors[3,19]. We demonstrate that WRN helicase inhibition results in preferential cell killing of multiple *BRCA2*-deficient cancer cell lines with distinct tissue origins, suggesting an avenue of exploration for therapy. Furthermore, mice administered the WRNi displayed elevated DNA damage and marked inhibition of *BRCA2*-deficient tumor growth in the xenograft model, in agreement with the hypersensitivity of *BRCA2*-deficient cancer cells to the WRNi. Drug-driven synthetic lethality, fueled by the seminal discovery of PARPi-induced selective cytotoxicity in *BRCA1*- or *BRCA2*-deficient cancer cells[64,65], represents a paradigm for developing therapeutics by exploiting tumor-specific genetic deficiencies[18,54,64–66]. Thus, our results provide a rationale for targeting *BRCA2* deficiencies in human cancers by pharmacological inactivation of DNA repair/replication stress response helicases implicated in fork stability maintenance. Moreover, WRN helicase inhibition markedly potentiated olaparib cytotoxicity in *BRCA2*-mutated ovarian cancer cells. This finding has a potentially important therapeutic implication as acquired resistance to PARP inhibitors is one of the major challenges in treating *BRCA2*-mutated ovarian cancers.

Based on our experimental findings, pharmacological inhibition of WRN helicase is a potential therapeutic strategy for tumors with *BRCA2* loss-of-function mutations. Recent works suggest that WRN is a druggable target in microsatellite-unstable human cancers[67–72]. Whether WRN's role at microsatellites directly relates to its function to stabilize replication forks under genetic conditions of fork deprotection remains to be seen. WRN and perhaps other replication stress response helicases represent potential targets in therapeutic strategies that rely on drug-induced or genetic synthetic lethality to combat cancer[72].

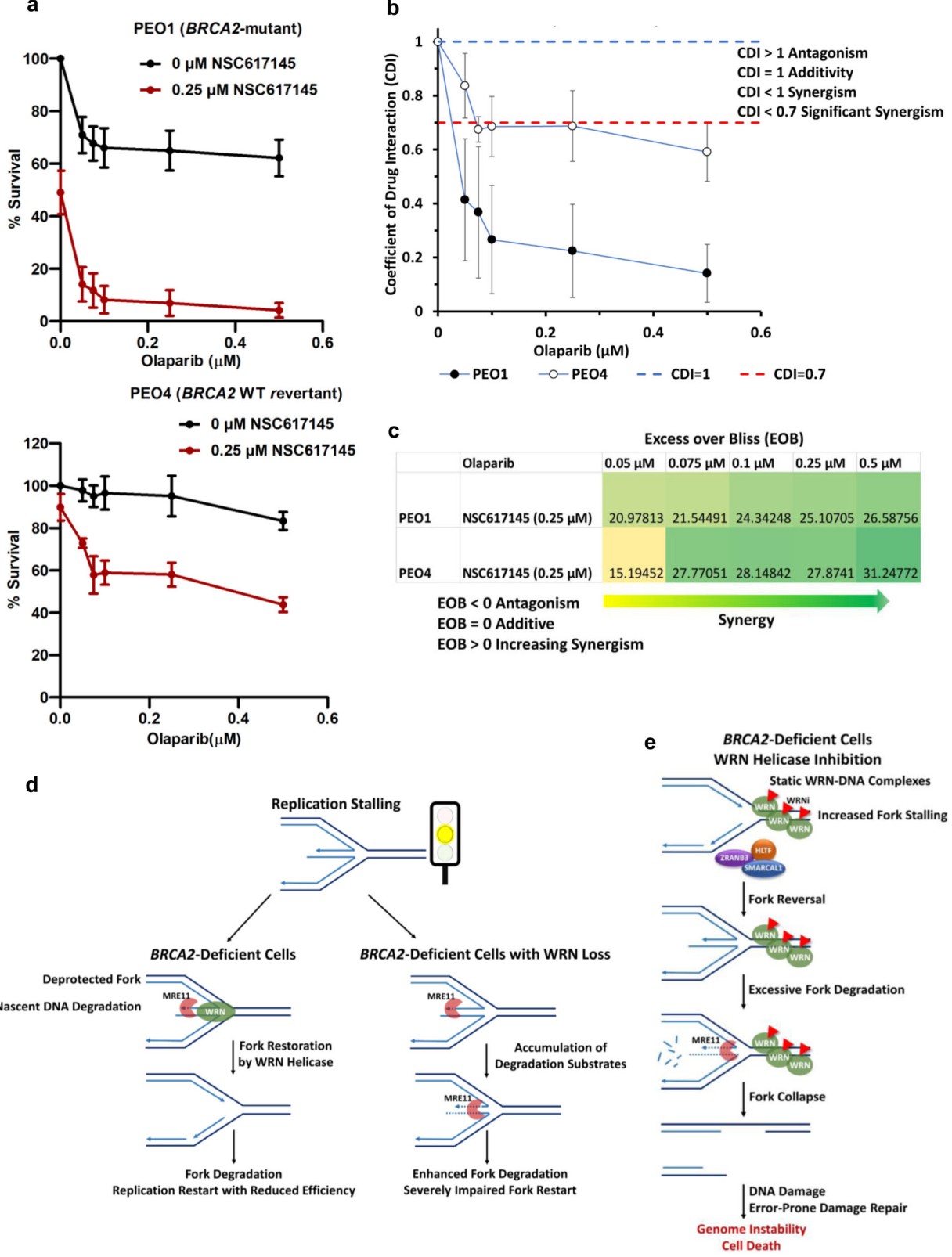

## Methods

**Cell culture, transfection, siRNAs, plasmids, recombinant proteins, and drug treatment**. DLD1 WT and isogenic *BRCA2* CRISPR KO cell lines were purchased from Horizon Discovery. PEO1 and PEO4 cells were provided by Dr. Lee Zou (Massachusetts General Hospital Cancer Center, Harvard Medical School). The PEO1 ovarian cancer cell line is characterized by a hemizygous nonsense mutation in *BRCA2* (c.5193C>G, Y1655X) resulting in an N-terminal truncated form of BRCA2 deficient in HR[37] and fork protection[35] functions. The PEO4 cell line was

derived from a cisplatin-refractory tumor of the same patient, in which the WT BRCA2 function was restored by a secondary mutation (c.5193C>T, Y1655Y)[37]. SKOV-3, MDA-MB-231, U2OS, and HeLa cells were purchased from ATCC. FANCD2$^{-/-}$ PD20 cells and corrected counterparts were obtained from Fanconi Anemia Cell Repository at Oregon Health & Science University (Portland, OR). Cell lines were cultured in RPMI-1640 medium (A1049101, Life Technologies) supplemented with 10% fetal bovine serum (Gibco) and 1% Pen Strep (Gibco) at 37 °C, 5% $CO_2$. For WRN gene KO, the commercially available WRN

**Fig. 9 WRNi potentiates olaparib cytotoxicity in *BRCA2*-mutated cancer cells. a** Relative survival (%) of colonies in PEO1 (upper panel) and PEO4 (lower panel) cells treated with either graded concentrations of olaparib only or in combination with 0.25 μM NSC617145. Cells were plated on a six-well plate at 1,000 cells/well and exposed to 0.25 μM WRNi and indicated doses of olaparib 4 h after cell seeding. Cells were allowed to form colonies in presence of the drugs for 12 days. Data represent mean ± SD of three (*n* = 3) independent experiments. **b** Coefficient of drug interaction (CDI) for indicated concentrations of olaparib and 0.25 μM NSC617145 as described in **a**. Dashed lines in the graph represent thresholds for synergism (blue) and significant synergism (red). Data represent mean ± SD of three (*n* = 3) independent experiments. **c** Excess over Bliss (EOB) scores obtained from olaparib-NSC617145 Bliss synergy analysis are shown with a color heat map. Survival (%) data from three independent experiments were used to calculate CDI and EOB in **b** and **c**, respectively. **d** and **e** Schematic model of defective replication restart and fork hyper-degradation induced by WRN loss (**d**) or targeted WRN helicase inhibition (**e**) in BRCA2-deficient cancer cells. **d** Model depicting the involvement of WRN in stabilizing replication forks under the condition of BRCA2 deficiency. BRCA2 deficiency renders reversed forks deprotected and vulnerable to MRE11 nuclease-mediated degradation. Following replication fork stalling, WRN helicase-mediated restoration of reversed forks to active replication forks limits MRE11-mediated uncontrolled nascent strand degradation and ensures fork restart, albeit with reduced efficiency. WRN loss results in increased availability of reversed fork substrates for MRE11-mediated degradation leading to enhanced nascent DNA degradation and severely compromised replication restart under BRCA2-deficient conditions. The yellow light of the traffic signal icon depicts replication stress. **e** Pharmacological inhibition of WRN helicase causes increased fork stalling induced by chromatin sequestration of WRN, followed by SNF2 translocase-mediated fork reversal and increased chromatin enrichment of MRE11 nuclease. In the absence of BRCA2, forks stalled by static WRN–DNA complexes are subjected to MRE11-mediated extensive degradation resulting in fork collapse, genomic instability, and cell death. Source data are provided as a Source Data file.

CRISPR/Cas9 KO plasmids (sc-401860, Santa Cruz Biotechnology) were used. Transfected cells were FACS-sorted into 96-well plates using a BD FACSAria II instrument. Resulting U2OS/*WRN*[−/−] or DLD1/*WRN*[−/−] colonies were screened by Western blotting.

HeLa/*RECQL1 KO* cells generated in this study were stably transfected with EV or WT RECQL1 in the lab of Dr. Grant Stewart (Institute for Cancer and Genomic Sciences, University of Birmingham, Birmingham, UK). In typical RNAi depletion experiments, cells (1 × 10[5]) grown in a six-well plate were transfected with either non-targeting control siRNAs or gene-specific siRNAs at a final concentration of 80 nM using DharmaFECT 1 (Dharmacon) or Lipofectamine 2000 (11668019, ThermoFisher Scientific). The following siRNAs were used: WRN (L-010378-00, ON-TARGETplus SMARTpool, Dharmacon), BRCA2 (L-003462-00, ON-TARGETplus SMARTpool, Dharmacon), RAD51 (L-003530-00, ON-TARGETplus SMARTpool, Dharmacon), SMARCAL1 (D-013058-04, siGenome SMARTpool, Dharmacon), HLTF (L-006448-00, ON-TARGETplus SMARTpool, Dharmacon), ZRANB3 (L-010025-01, ON-TARGETplus SMARTpool, Dharmacon), RAD52 (L-011760-00-0005, ON-TARGETplus SMARTpool, Dharmacon), MUS81 (L-016143-01-0005, ON-TARGETplus SMARTpool, Dharmacon) and non-targeting control siRNAs (D-001810-01, ON-TARGETplus, Dharmacon). For fork stability experiments, U2OS/*WRN*[−/−] or DLD1/*WRN*[−/−] cells (1 × 10[5]) were transiently transfected with 2 μg of pCMV-Flag- WT-WRN[24] or WRN site-directed mutant plasmids using Lipofectamine 2000 and subjected to the fork stability assay 48 h post-transfection as described below. To assess the effects of RECQL1 overexpression on WRNi-induced fork degradation, PEO1 cells (1 × 10[5]) were transiently transfected with 2 μg of either EV (pCMV6-Entry, PS100001, Origene) or RECQL1 plasmid (pCMV6-RECQL1, RC200427, Origene) using Lipofectamine 2000 and subjected to the fork stability assay 48 h post-transfection. All transfections were carried out in serum and antibiotic-free RPMI-1640 medium.

Recombinant FLAG-WRN proteins were expressed in insect cells. FLAG-WRN was subcloned from the pFLAGWRN plasmid (WRN CDK1[24]) into the pFastBacHTB construct (Thermo Fisher Scientific) via 5′ NotI and 3′ KpnI sites, making the pFastBacHTB-FLAG-WRN plasmid. Site-directed mutagenesis was carried out by Genewiz, Inc. to generate pFastBacHTB-FLAG-WRNE84A, pFastBacHTB-FLAG-WRNK577M and pFastBacHTB-FLAG-WRNE84A/K577M plasmids. All plasmids were sequenced to confirm their identity. Miniprep plasmids were transformed into DH10Bac to generate baculovirus DNA which was transfected into 1 × 10[6] Sf9 insect cells using Cellfectin II transfection reagent (Thermo Fisher Scientific) according to the manufacturer's protocol. Five days post-transfection, the media containing the baculovirus was isolated and used to reinfect Sf9 insect cells for 5 additional days generating a higher titer virus. Media containing the baculovirus was stored at 4 °C. High titer baculovirus for pFastBacHTB-FLAG-WRN, pFastBacHTB-FLAG-WRNE84A, pFastBacHTB-FLAG-WRNK577M, and pFastBacHTB-FLAG-WRNE84A/K577M were used to infect Hi5 insect cells (Thermo Fisher Scientific) at an approximate MOI of 10. Hi5 cells were harvested 48 h post-infection by washing in 1× PBS with Complete Ultra Protease Inhibitor Cocktail (Roche) and centrifugation at 1500 RPM for 5 min and pellets placed in −80 °C until lysed. Cell pellets containing approximately 1 × 10[8] cells were resuspended in 10 ml of Lysis Buffer (50 mM Tris pH 7.4, 150 mM NaCl, 0.4% NP40, 10% glycerol, 5 mM BME, and Complete Ultra Protease Inhibitors (Roche)) and rotated at 4 °C for 45 min. Lysates were centrifuged at 20,000 RPM for 10 min. Supernatants were filtered through a 0.45 μm PVDF filter. Each clarified lysate was passed twice through a Ni2+-charged 1 ml HiTrap Chelating HP column (Cytiva) equilibrated in Buffer TN (50 mM Tris pH 7.4, 150 mM NaCl, 10% glycerol, 5 mM BME, protease inhibitors) with 10 mM imidazole. Three washes with TN buffer containing 10, 20, and 40 mM imidazole were performed followed by elution with TN buffer containing 400 mM imidazole. Eluted protein was pooled and digested with TEV protease for 16 h at 4 °C to remove the 6×His tag. The protein was dialyzed into NETN-500 Buffer (50 mM Tris pH 7.4, 500 mM

NaCl, 0.5% NP40, 1 mM EDTA) using an Amicon Ultra 100 kD cutoff centrifugal filter (EMD Millipore). The retained sample was applied to 250 μl of packed M2 anti-FLAG beads (Sigma) which had been equilibrated in NETN-500 buffer. The beads were washed twice with 5 ml of NETN-500 buffer and the protein was eluted twice in 500 μl Storage Buffer (100 mM Tris pH 8.0, 400 mM NaCl, 10% glycerol, 5 mM BME) containing 3× FLAG peptide. Eluted protein was concentrated and dialyzed against Storage Buffer without 3X FLAG peptide and frozen in liquid nitrogen followed by storage at −80 °C. Recombinant MRE11 was provided by Dr. Jean-Yves Masson, Laval University Cancer Research Center, Québec City, Canada. Treatment with NSC617145 (5340, Tocris), olaparib (S1060, Selleckchem), HU (H8627, Millipore Sigma), Mirin (M9948, Millipore Sigma), C5 (HY-128729, MedChemExpress), and NU7441 (KU-57788, S2638, Selleckchem) was performed in complete growth medium at 37 °C with 5% $CO_2$.

**Oligonucleotide substrate preparations.** The regressed fork, i.e., Chicken-foot (CF) substrate, and replication fork (RF) substrate were prepared using oligonucleotides listed in Supplementary Table 1 as follows: For fork restoration assays, 10 pmol of oligo B was labeled at its 5′ end using T4 polynucleotide kinase and 30 μCi γ[32]P-ATP in a 20 ul reaction at 37 °C for 40 min. After heat inactivation at 65 °C for 20 min, free ATP was removed by applying the sample to a G-25 spin column (Cytiva). 10 pmol of either oligo A (for CF) or oligo D (for RF) was annealed to the labeled oligo B in 50 μl total volume with 10 mM Tris–HCl, pH 7.5 and 50 mM NaCl by heating the oligonucleotides at 95 °C for 5 min followed by slow cooling to RT. 50 pmol of Oligo C and either oligo D (for CF) or A (for RF) were annealed similarly in 10 mM Tris, pH 7.5, and 50 mM NaCl. Just prior to the fork restoration assay, the CF substrate was prepared by combining 1 pmol of radiolabeled substrate AB and 2 pmol of unlabeled substrate CD in annealing buffer (10 mM Tris–HCl pH 7.5, 10 mM MgCl2, 50 mM NaCl) and incubating at 37 °C for 30 min followed by RT for an additional 30 min. For the replication fork, radiolabeled 1 pmol of BD substrate was annealed to 2 pmol of substrate AC in a similar fashion.

**In vitro fork restoration assay.** The indicated concentrations of WRN, WRN-E84A, WRN-K577M, or WRN-E84A/K577M were incubated with the CF substrate (2 nM) in 20 μl reactions containing branch migration reaction salts (35 mM Tris–HCl, pH 7.5, 20 mM KCl, 5 mM MgCl2, 0.1 mg/ml BSA, 2 mM DTT, 15 mM phosphocreatine, 30 U/ml creatine phosphokinase and 5% glycerol) and 2 mM ATP for 20 min at 37 °C. Ten μl of 3× stop solution (1.2% SDS, 30% glycerol, and 0.8 U proteinase K) was added to each reaction for 10 min at RT. Samples were immediately loaded onto an 8% 1× TBE native PAGE gel electrophoresed at 200 V for 4 h at 4 °C. Gels were exposed to a phosphorscreen overnight and imaged using a Typhoon FLA 9500 (Cytiva).

**Electrophoretic mobility shift assay.** The indicated concentrations of WRN, E84A-WRN, K577M-WRN, E84A/K577M-WRN were incubated with the CF substrate (2 nM) in 20 μl reactions containing branch migration reaction salts (35 mM Tris–HCl, pH 7.5, 20 mM KCl, 5 mM MgCl2, 0.1 mg/ml BSA, 2 mM DTT, and 5% glycerol) and 2 mM ATPγS for 15 min at RT. 4 μl of loading dye (25% glycerol, 0.02% bromophenol blue, and 0.02% xylene cyanol) were added to each reaction and immediately loaded onto a 5% 1× TBE native PAGE gel which was electrophoresed at 200 V for 2 h at 4 °C.

**In vitro nuclease assay.** CF substrate was prepared as described above, however oligonucleotide C was radiolabeled and annealed to oligonucleotide D and oligonucleotides A and B were annealed, followed by annealing of the partial duplex CD to partial duplex AB. The indicated concentrations of WRN-E84A were incubated

with the CF substrate (0.5 nM) in 10 μl reactions containing modified branch migration reaction salts (35 mM Tris–HCl, pH 7.5, 5 mM MnCl₂, 1 mM MgCl₂, 0.1 mg/ml BSA, 2 mM DTT, 15 mM phosphocreatine, 30 U/ml creatine phospho-kinase and 5% glycerol) and 2 mM ATP for 10 min at 37 °C. After the initial reaction, MRE11 was added to the reaction at a final concentration of 5 nM and incubated for an additional 50 min at 37 °C. One microliter of proteinase K (0.8 U) was added to the reaction mixture and incubated at 37 °C for 15 min followed by the addition of 10 μl of formamide loading buffer (80% formamide, 0.1% xylene cyanol, 0.1% bromophenol blue, 0.5× TBE). Samples were heated at 95 °C for 5 min and then loaded on 12% denaturing polyacrylamide gels with 8 M urea and 0.5× TBE. Gels were electrophoresed for 40 min at 50 W and exposed to a phos-phorscreen overnight and imaged using a Typhoon FLA 9500 (Cytiva). Percent exonuclease activity was determined by taking the volume of bands below the intact oligo C for each lane of the gel and dividing by the total volume of the lane using ImageQuant TL (Cytiva).

**Western blotting**. Total cellular protein was extracted using RIPA cell lysis buffer (89900, Pierce) supplemented with protease inhibitor cocktail (78429, Thermo-Fisher Scientific) followed by sonication. Protein concentration was determined using BCA protein assay kit (23225, Pierce), and 30–50 μg of total proteins were either resolved by electrophoresis on NuPAGE™ 4–12% Bis–Tris (NP0335BOX, Ther-moFisher Scientific) or NuPAGE™ 3-8% Tris–Acetate (EA0378BOX, ThermoFisher Scientific) polyacrylamide gels. Proteins were transferred onto 0.45 μm PVDF membrane (IPVH00010, Millipore Sigma) by cold wet transfer method using NuPAGE™ Transfer buffer (NP0006-1, ThermoFisher Scientific). The membranes were blocked with 5% BSA (1073508600, Sigma Millipore) followed by incubation with primary antibodies overnight at 4 °C. The following primary antibodies were used: anti-WRN (4666, Mouse mAb (8H3), Cell Signaling, 1:1000), anti-BRCA2 (A303-434A-M, Bethyl Laboratories, 1:1000; D9S6V, 10741, Cell Signaling, 1:1000), anti-RAD51 (ab63801, Abcam, 1:1000), anti-SMARCAL1 (A301-616A-T, Bethyl Laboratories, 1:1000), anti-ZRANB3 (A303-033A-T, Bethyl Laboratories, 1:1000), anti-HLTF (A300-230A-T, Bethyl Laboratories, 1:1000), anti-β-actin (12620, D6A8, Cell Signaling, 1:5000), anti-β-tubulin (N-20, sc-9935, Santa Cruz Biotechnology, Inc., 1:2000), anti-DNA2 (ab96488, Abcam, 1:1000), anti-Mus81 (MTA30 2G10/3, ab14387, Abcam, 1:1000), anti-Rad52 (F-7, sc-365341, Santa Cruz Biotechnology Inc., 1:1000), anti-Exonuclease 1 (ab95068, Abcam, 1:1000), anti-MRE11 (12D7, ab214, Abcam, 1:1000), anti-RECQL1 (H-110, sc-25547, Santa Cruz Biotechnology Inc., 1:1000), anti-Topoisomerase I (556597, BD Pharmigen, 1:500), anti-Lamin B1 (D9V6H, 13435, Cell Signaling, 1:1000), anti-γ-H2AX (JBW301,05-636, Millipore Sigma, 1:1000), and anti-ORC2 (A302-734A, Bethyl Laboratories, 1:1000). The membranes were incubated with HRP-conjugated anti-mouse (32430, ThermoFisher Scientific, 1:2500 dilution), anti-rabbit (656120, ThermoFisher Scientific, 1:2500 dilution) or anti-goat (SC-2020, Santa Cruz Bio-technology Inc., 1:2500 dilution) secondary antibodies at room temperature (RT) for 2 h. After washing with 1× PBST (Phosphate Buffer Saline with 0.1% Tween 20), the membranes were subjected to ChemiDoc imaging system (Biorad) and protein bands were detected using SuperSignal™ West Pico PLUS Chemilumines-cent (34580, ThermoFisher Scientific) or Luminata Forte Western HRP (WBLUF0100, Millipore Sigma) substrate. Quantification of Western blots was done using Image lab 6.0.1 software (Biorad).

**DNA fiber assay**. Asynchronously growing cells (~0.1–0.3 × 10⁶) were sequentially labeled with 20 μM CldU (C6891, Millipore Sigma) and 200 μM IdU (I7125, Millipore Sigma) thymidine analogs for time duration as indicated in the figure panels of the respective fiber experiments. Drug doses and incubation time for individual fiber experiments are indicated in the fiber labeling scheme of the respective figures. Cells were washed twice with ice-cold 1× PBS, trypsinized, and approximately 2000 cells were spotted and lysed (200 mM Tris–HCl pH 7.4, 50 mM EDTA, and 0.5% SDS) on silane-coated slides (5070, Newcomer Supply). The DNA fibers were stretched along with the slide by gravity followed by air-drying and fixation (Methanol:Acetic acid, 3:1). DNA fibers were denatured by treating with 2.5 M HCl at RT for 1 h. Slides were neutralized with 400 mM Tris–HCl, pH 7.4, washed with 1× PBST and blocked in 5% BSA and 10% goat serum at 4 °C overnight. The slides were subsequently incubated with rat anti-CIdU antibody (1:200, ab6326, Abcam) for 1 h at RT. After three washes with 1× PBST, the slides were incubated with AlexaFluor 647-conjugated anti-rat IgG secondary antibody (1:100, A-21247, ThermoFisher Scientific) for 1 h at RT fol-lowed by stringent PBST wash. In a similar way, the slides were further incubated with mouse anti-IdU antibody (1:40, 347580, BD Pharmigen) and AlexaFluor 488-conjugated anti-mouse IgG secondary antibody (1:100, A-11001, ThermoFisher Scientific) for 1 h each. Finally, the slides were mounted with Prolong Gold antifade mountant (P36930, ThermoFisher Scientific). Labeled DNA fibers were examined using Zeiss Axio Observer.Z1/7 fluorescence microscope with a 63x (oil) objective lens. Cy5 and GFP filters were used for the detection of CldU and IdU labeled fibers, respectively. The fibers were analyzed using ZEN 3.0 (blue edition) software.

To measure the fork stability upon genetic knockdown, cells (1 × 10⁵) were transfected with specific siRNAs 72 h prior to fiber labeling. U2OS and DLD1 WRN KO cells transfected with control or BRCA2 siRNA (80 nM) were re-transfected with 2 μg of empty vector (EV), WT WRN, or catalytic dead WRN

mutant plasmids 24 h post-siRNA transfection. Transfected cells were subjected to fork stability assay 48 h after plasmid transfection. Following CldU and IdU labeling, cells were washed twice with warm 1× PBS and subsequently incubated with 4 mM HU for 5 h before harvesting. To assess the effect of WRN or DNA2 inhibition on the stability of HU-stalled forks in PEO1 and PEO4 cells, WRNi NSC617145 (4 μM) or DNA2 inhibitor C5 (25 μM) was added to the culture medium concomitantly with 4 mM HU after the IdU labeling. DLD1 cells were treated with WRNi (1 μM) for 1 h and subsequently labeled with CldU and IdU nucleotides before the addition of HU. For assessing the effect of NSC617145 alone on fork stability, PEO1 and PEO4 cells were incubated with 4 μM of NSC617145 for 5 h following sequential labeling with CldU and IdU. To assess the effect of MRE11 or DNA2 inhibition on NSC617145-induced fork degradation, PEO1 and PEO4 cells were optionally pretreated with 50 μM Mirin or 25 μM C5 for 1 h prior to nucleotide labeling and the inhibitors were present throughout the experiment. DNA fibers with contiguous CldU–IdU signals were scored and the length of CldU and IdU tracts of individual fibers were measured manually using ZEN 3.0 (blue edition) software. IdU to CldU tract length ratios (IdU/CldU) of individual DNA fibers were calculated in Excel spreadsheet and scatter dot plots were generated using GraphPad Prism version 5.03.

To assess fork restart efficiency in BRCA2-depleted cells upon WRN complementation, U2OS WRN KO cells (1 × 10⁵) were transfected with either control or BRCA2 siRNA (80 nM). Twenty-four hours post-siRNA transfection, cells were re-transfected with 2 μg of empty vector (EV), WT WRN, or catalytic dead WRN mutant plasmids. Transfected cells were subjected to fork restart assay 48 h after plasmid transfection. Cells were treated with 2 mM HU during CldU labeling to induce replication stalling. After 60 min incubation with HU, cells were washed three times in warm 1× PBS and incubated further in HU-free medium containing IdU for 45 min. For the fork restart experiments in PEO1 and PEO4 cells, control or WRN siRNA (80 nM) transfected cells were subjected to fork restart assay 72 h post-siRNA transfection. After 60 min incubation with 2 mM HU in the presence of CldU, cells were washed three times in warm 1× PBS and incubated in an HU-free medium containing IdU nucleotides for 60 min. In experiments with the WRNi, PEO4 and PEO1 cells were cotreated with 1 μM NSC617145 and 2 mM HU concomitantly with CldU and IdU labeling. IdU tract lengths of the restarted forks (forks characterized by contiguous CldU–IdU label) were measured to assess fork restart efficiency under different experimental conditions. To negate the possible effect of fork progression defect caused by different experimental conditions on restart efficiency quantitation, the length of an IdU tract was normalized to that of the corresponding CldU tract for a given fork. To analyze stalled fork recovery upon RAD52 knockdown in BRCA2-mutated cells exposed to WRNi, PEO1 cells transfected with either control or RAD52 siRNA (80 nM) were subjected to fork restart assay 72 h post-transfection. After CldU labeling for 20 min, cells were incubated in 2 mM HU for 4 h. After PBS wash, cells were subsequently incubated in HU-free medium containing IdU nucleotides for 20 min. WRNi (1 μM) was added to the medium during CldU labeling and present throughout the experiment. The percentage of stalled forks was determined as a percentage of only red-labeled fibers among the total red labeled (only red + red-green fibers) DNA tracts.

For fork progression analysis, exponentially growing cells (1 × 10⁵) were labeled with CldU for 15 min followed by IdU labeling for 60 min. To determine the immediate effect of WRNi on fork progression, cells were treated with 2 μM NSC617145 concomitantly with the IdU labeling. IdU to CldU tract length ratios of individual ongoing forks (fork with continuous CldU–IdU signal) were determined to assess relative fork progression upon WRN helicase inhibition. To determine fork asymmetry upon WRN helicase inhibition, sister tract (IdU) lengths of individual bi-directional fork structures (IdU–CldU–IdU) were measured, and short to long SF ratios were calculated. The relative decrease in the SF ratio serves as a readout of fork asymmetry.

For all fiber experiments, at least 125–300 untangled fibers were scored for each experimental condition. Statistical significance was determined by the Mann–Whitney test.

**Immunofluorescence**. Cells (40,000 cells per well) grown on four-well cell culture chamber slides (154526, ThermoFisher Scientific) were treated with either NSC617145 or DMSO control for the time duration as indicated in the figure legends of the respective immunofluorescence experiments. Cells were washed twice with ice-cold 1× PBS and nuclear pre-extraction was performed on ice using 500 μl of extraction buffer 1 (10 mM PIPES, pH 7.0; 100 mM NaCl; 300 mM Sucrose; 3 mM MgCl₂; 1 mM EGTA, 0.5% Triton X-100) and buffer 2 (10 mM Tris–HCl, pH 7.5; 10 mM NaCl; 3 mM MgCl₂, 1% Tween 40, 0.5% sodium deoxycholate) for 5 min each. Cells were washed with 1× PBS and fixed with 500 μl of 4% paraformaldehyde solution (sc-281692, SCBT) at RT for 10 min. For PCNA co-staining, cells were fixed with ice-cold Methanol at −20 °C for 20 min following paraformaldehyde fixation. Fixed cells were washed twice with 1× PBS and sub-sequently permeabilized with 0.5% Triton X-100 at RT for 5 min, followed by incubation in blocking buffer (5% BSA + 10% goat serum) at RT for 1 h. Cells were incubated with the following primary antibodies overnight at 4 °C: anti-γ-H2AX (JBW301,05-636, Mouse mAb, Millipore Sigma, 1:100), anti-γ-H2AX (20E3, Rabbit mAb #9718, Cell Signaling, 1:100), anti-53BP1 (clone BP13, MAB3802, Millipore

Sigma, 1:100), anti-DNA PKcs phospho-T2609 (ab18356, Abcam, 1:100), anti-DNA PKcs phospho-S2056 (EPR5670, ab124918, Abcam, 1:100), PCNA (PC10, sc-56, Santa Cruz Biotechnology Inc., 1:50), and anti-WRN (NB100-471, Novus Biologicals, 1:50). Slides were incubated with AlexaFluor 647-conjugated goat anti-mouse IgG (A-21235, ThermoFisher Scientific, 1:500 dilution for γ-H2AX (JBW301, Mouse mAb), 53BP1 and PCNA), AlexaFluor 488-conjugated goat anti-mouse IgG (A28175, ThermoFisher Scientific, 1:500 dilution for anti-DNA PKcs phospho-T2609) or AlexaFluor Plus 488 Goat anti-Rabbit IgG (A32731, ThermoFisher Scientific, 1:500 dilution for γ-H2AX (20E3, Rabbit mAb), DNA PKcs phospho-S2056 (EPR5670, Rabbit mAb) and WRN) secondary antibodies for 2 h at RT followed by three washes with 1X PBST. The slides were mounted with Pro-Long Diamond Antifade Mountant with DAPI (P36962, ThermoFisher Scientific) or VECTASHIELD® Antifade Mounting Medium with DAPI (H-1200-10, Vector Laboratories) and examined under Zeiss Axio Observer.Z1/7 fluorescence microscope with a 63x (oil) objective lens using ZEN 3.0 (blue edition) software. Analysis of immunofluorescence images was done using ZEN 3.0 (blue edition) software.

**Chromatin isolation**. Chromatin fractionation with CSK buffer containing 0.5% Triton X-100 was done as described previously[73] with some modifications. Approximately $2 \times 10^7$ cells grown in 150 mm dish were washed twice with 1× ice-cold PBS and collected by trypsinization. Cells were lysed in 1 ml modified CSK buffer (10 mM PIPES pH 6.8, 175 mM NaCl, 300 mM sucrose, 3 mM MgCl$_2$, 1 mM EGTA, 1 mM DTT, 0.5% Triton X-100) supplemented with 1X protease inhibitor cocktail (78429, ThermoFisher Scientific) for 10 min at 0 °C. Lysates were centrifuged at 5000×g for 3 min and the supernatant was collected as Triton-soluble fraction (cytosol + nucleosol). The pellet containing insoluble chromatin fractions were washed three times with CSK buffer each for 30 min on a rotator at 4 °C. The final pellet was resuspended in 100 μl of CSK buffer (without Triton X-100) supplemented with 1 μl/ml benzonase (E1014, Millipore Sigma) and incubated for 15 min at 37 °C. 1× LDS sample buffer (NP0007, ThermoFisher Scientific) was added to the sample and sonicated for 20 s using a probe sonicator (Branson digital sonifier) with 10% amplitude. Samples were clarified by centrifugation at 18,000×g for 10 min and supernatants were collected as chromatin-bound fractions.

**Radiolabeling of NSC617145 and <sup>14</sup>C-NSC617145-WRN binding assay**. Radiolabeling of NSC617145 was performed under contract by Moravek, Inc. In brief, a condensation reaction was carried out, whereby 2,3-dichloromaleic anhydride was reacted with $^{14}$C-labeled 2,2,-dimethyl-1,3-propanediamine in acetic acid. The solid product was isolated by filtration and subsequently stored as a 0.1 mCi/mL solution in acetonitrile. Radiochemical purity was 98%.

For the in vitro $^{14}$C-NSC617145-WRN binding assay, an equivalent concentration (300 nM) of purified recombinant WRN, FANCJ, or RECQL1 protein was incubated with 10 μM $^{14}$C-NSC617145 in 20 μl binding buffer (100 mM Tris pH 8.0, 400 mM NaCl) for 15 min at 37 °C. Using a 96-well Bio-dot microfiltration apparatus (#170-6545, Bio-Rad), the binding incubation samples were applied onto 0.2 μm nitrocellulose membrane pre-soaked in 1× TBS (Tris Buffered Saline) and allowed to filter through the membrane by gravity flow. The membrane was washed three times with 1× TBS by applying the vacuum before exposing it to a phosphorscreen. $^{14}$C signal was detected with Typhoon FLA 9500 (GE Healthcare) and quantified using ImageQuant software (GE Healthcare). To determine $^{14}$C-NSC617145-WRN binding in vivo, $WRN^{-/-}$ U2OS cells ($\sim 8 \times 10^6$) were transfected with 30 μg of either FLAG-empty vector or FLAG-WRN expression plasmids using Lipofectamine 2000 (11668019, ThermoFisher Scientific). Forty-eight hours post-transfection, cells were treated with 10 μM $^{14}$C-NSC617145 for 1 h and subsequently lysed in 1 ml lysis buffer (50 mM Tris–HCl pH 7.4, 150 mM NaCl, 1 mM EDTA, and 1% Triton X-100) complemented with benzonase (1 μl/ml) and protease inhibitor cocktail. Cell lysates were centrifuged for 10 min at 12,000×g at 4 °C and the supernatant was incubated with 25 μl of anti-FLAG M2 affinity beads (A2220, Millipore Sigma) overnight at 4 °C with gentle rotation. Beads were washed three times with lysis buffer and $^{14}$C-NSC617145: FLAG-WRN complexes were eluted (twice) by incubating beads with 100 μl of 3× FLAG peptide (250 ng/μl final concentration, F4799, Millipore Sigma) for 45 min on a rotator at 4 °C. Eluted samples were subjected to dot blot assay using 96-well Bio-dot microfiltration apparatus as described above.

**Neutral comet assay**. DSBs upon WRN helicase inhibition were analyzed using neutral comet assay (single-cell gel electrophoresis) according to the Comet SCGE assay kit (Enzo, ADI-900-166) protocol. PEO4 and PEO1 cells ($1 \times 10^5$ cells) were transfected with either control or MUS81 siRNA (80 nM) in a six-well plate. Seventy-two hours post-transfection, cells were treated with WRNi (5 μM) for 6 h. Cells were collected by trypsinization, counted, and resuspended in ice-cold 1× PBS (Ca$^{2+}$ and Mg$^{2+}$ free) at $1 \times 10^5$ cells/ml. Cells were mixed with molten low-melting agarose (at 37 °C) at a ratio of 1:10 (v/v) and 75 μl of agarose/cells suspension was immediately spread on the sample area of CometSlides (Enzo). Slides were kept at 4 °C in the dark until the agarose solidified. Next, slides were immersed in prechilled 1× lysis buffer (Enzo) at 4 °C for 1 h. Slides were washed once in 1× Tris–Borate–EDTA (TBE) buffer and subjected to horizontal gel electrophoresis in ice-cold 1× TBE buffer with 21 V for 20 min at 4 °C. Slides were briefly rinsed in deionized water, immersed in 70% ethanol for 10 min, and kept at

37 °C for drying. Samples in dried agarose were stained with CYGREEN nucleic acid dye (Enzo, diluted at a ratio of 1:1000) for 30 min at RT in the dark. Slides were briefly rinsed in deionized water and mounted using Permount mounting medium (Fisher Scientific). Comet slides were visualized using a GFP filter of Zeiss Axio Observer.Z1 fluorescence microscope with 10x objective lens. Comet tail moment was analyzed using OpenComet analysis tool[74] of ImageJ. Statistical analysis was performed in GraphPad Prism 5, version 5.03.

**Metaphase spread preparation**. PEO1 and PEO4 cells ($2.5 \times 10^6$) grown in 100 mm dishes were treated with 2 μM NSC617145 for 48 h. To assess the effects of MRE11 or DNA-PK inhibition on WRNi-induced chromosomal instability in $BRCA2$-mutated cells, PEO1 cells ($5 \times 10^6$) were treated with WRNi (5 μM) for 16 h in the presence or absence of Mirin (25 μM) or NU7441 (10 μM), respectively. Cells ($\sim 8 \times 10^6$) were arrested in metaphase by incubating with KaryoMAX Colcemid solution (0.1 μg/ml) (15212012, ThermoFisher Scientific) for 3 h, washed twice with 1× ice-cold PBS, trypsinized, and collected in 15 ml tubes by centrifugation at 100×g for 5 min. Cell pellets were gently resuspended in 1 ml ice-cold 1× PBS followed by dropwise addition of 9 ml preheated (37 °C) KCl (75 mM) solution while slowly vortexing the tubes. After 20 min incubation in KCl solution at 37 °C, cells were pelleted by centrifugation at 100×g for 5 min. Cells were fixed (two times) on ice by adding 2 ml freshly prepared fixative solution (methanol:acetic acid, 3:1) dropwise with mild vortexing. After adding another 4 ml of fixative solution, cells were pelleted by centrifugation at 100×g for 5 min. Cell pellets were resuspended in 1 ml fixative solution and kept at −20 °C overnight. The chromosome spreads were prepared by dropping 200 μl of the cell suspension on 40% acetic acid spread on a glass slide (5070, Newcomer Supply). The slides were air-dried and mounted with Prolong Gold antifade mountant (P36930, ThermoFisher Scientific). The slides were visualized under Zeiss Axio Observer.Z1/7 microscope with a 6x3 (oil) objective lens and images of metaphase chromosomes were acquired with the color camera attached. Chromosomal aberrations were analyzed using ZEN 3.0 (blue edition) software and graphs were generated in GraphPad Prism, version 5.03.

**Colony formation assay**. Cells ($1 \times 10^5$) were transfected with siRNAs (80 nM) and plated (500–1,500 cells) on 6-well, 12-well or 24-well tissue culture dishes 72 h post-transfection. Drugs were added to the medium 4 h after cell seeding and cells were grown to form colonies in presence of drugs for 10–16 days. The number of cells plated, and colony formation times used for individual cell lines are detailed in the respective figure legends. Colonies were washed twice with 1× PBS, fixed with 100% methanol, and stained with crystal violet staining solution (25% methanol + 0.5% crystal violet). Percent survival upon drug treatment was determined by normalizing the total number of colonies formed with plating efficiency under individual conditions. Graphs were generated using GraphPad Prism, version 5.03.

**WST1 cytotoxicity assay**. Cell viability in response to WRNi treatment was assessed using WST1 cell viability assay reagent (1164480700, Millipore Sigma) according to the manufacturer's protocol. Cells (1,500 cells in 100 μl complete cell culture medium) were plated on a 96-well plate and grown overnight before the addition of WRNi. Drug incubation times for individual experiments are given in the respective figure legends. For WST experiments following siRNA transfection, cells were first transfected with specific siRNAs (80 nM) in a six-well plate and subsequently plated on a 96-well plate (1500 cells) 24 h post-transfection. After 48 h of siRNA transfection, cells were exposed to WRNi for another 48 h before the addition of WST1 reagent (diluted 1:10 in cell culture medium). After 1 h incubation at 37 °C, the absorbance of the samples was taken with an ELISA microplate reader at 450 nm. Relative cell viability upon WRNi treatment was determined with respect to untreated cells and line graphs were generated in Excel or GraphPad Prism 5, version 5.03.

**In vivo mouse xenograft experiments**. Mice were maintained as outlined in the Guide for the Care and Use of Laboratory Animals, under a protocol approved by the Animal Care and Use Committee (ACUC) of NCI-Frederick.

BRCA2-deficient and BRCA2-proficient DLD1 human colorectal adenocarcinoma cells were injected subcutaneously into the flanks of 8-week-old female athymic nude (NCr-nu/nu) mice (NCI-Frederick animal facility) at a concentration of $5 \times 10^6$ in 200 μl of PBS. The following number of mice were used for four groups: DLD1 $BRCA2^{+/+}$ treated with vehicle, $n = 7$; DLD1 $BRCA2^{-/-}$ treated with vehicle, $n = 7$; DLD1 $BRCA2+/+$ treated with NSC617145, $n = 7$; and DLD1 $BRCA2^{-/-}$ treated with NSC617145, $n = 6$. The resource equation method was used to determine sample sizes. No selection criteria were used for selecting the mice for each group. After the tumor volume reached 100–150 mm$^3$, vehicle or NSC617145 at a dose of 20 mg/kg was administered by IP injection every other day for 2 weeks (total 7 treatments). WRNi NSC617145 was dissolved (1.25 mg/ml) in 3% DMA (N, N-dimethylacetamide), 1.8% Tween 80 in D5W (5% Dextrose in water) for IP injection. Tumor size was measured every other day for 3–4 weeks. Tumors were harvested before they reach the maximal size (2000 mm$^3$) permitted by NCI-Frederick Animal Care and Use Committee (ACUC). After harvesting, tumors were flash-frozen in liquid nitrogen for further characterization. Tumor sizes were measured using a digital Vernier caliper and tumor volume (in mm$^3$)

was calculated as a product of (length × width$^2$)/2. Tumor growth inhibition (TGI) was calculated using the formula, % TGI = (1−(mean volume of treated tumors)/(mean volume of control tumors))×100. Mice were maintained according to the procedures outlined in the Guide for the Care and Use of Laboratory Animals and a protocol approved by the NCI-Frederick Animal Care and Use Committee (ACUC). Mice were housed as per NCI-Frederick ACUC guidelines as described below: Housing: All mice were group-housed unless scientific justification was approved by the NCI-Frederick ACUC (e.g., the last animal left in a study group). All cages had at least one form of environmental enrichment. All singly housed mice were provided at least two forms of enrichment, such as a plastic hut or additional type of nesting material; Light cycle: Rooms were maintained at a constant light cycle of 12 h on and 12 h off; Temperature: Rooms were maintained at temperatures in the range of 68–79 °F; Humidity: Relative-humidity levels were maintained at 30–70%. Animal caretakers checked room humidity by using dedicated digital thermometer/hygrometers during the AM and PM animal and cage observations; Air changes: air exchanges in the animal rooms were in the range of 10–15 air changes per hour (ACH).

**Immunohistochemistry (IHC)**. IHC of mouse xenograft tumor tissues was carried out at Pathology/Histotechnology Laboratory (PHL), Frederick National Laboratory for Cancer Research. Immunohistochemical staining was performed on the Bond RX autostainer (LeicaBiosystems). Following antigen retrieval with Bond ER1, slides were stained for γH2AX using the Bond Intense R Detection kit (DS9263, LeicaBiosystems) and 60 min incubation of biotinylated phospho-Histone H2AX (#16-193, Millipore, diluted 1:100). Slides for 53BP1 were subjected to Bond ER2 antigen retrieval and stained using the Bond Polymer Refine Detection kit (DS9800, LeicaBiosystems) and 30 min incubation of 53BP1 (EPR2172(2), ab175933, Abcam, diluted 1:200). DAB (3,3′ diaminobenzidine) was used as chromogen. IHC slides were photographed at 20x and 40x magnification using a color camera attached to Zeiss Axio Observer.Z1/7. Integrated density (IntDen) of γH2AX or 53BP1 was quantified using Image-J (https://imagej.nih.gov/ij/) and box plots were generated in GraphPad Prism, version 5.03.

**Drug interaction analysis**. The CDI was determined using the following equation: CDI = AB/(A × B) where AB is the ratio of the viability of cells treated with both NSC617145 and Olaparib to the no treatment cells; A (NSC617145) or B (Olaparib) is the ratio of the cell viability of the cells treated with either NSC617145 or Olaparib to the no treatment cells. A CDI values can be separated into several categories: CDI > 1 (antagonistic), CDI = 1 (additive), CDI < 1 (synergistic), or CDI < 0.7 (significantly synergistic)[75]. CDI values were calculated at different concentrations of Olaparib with a single concentration of NSC617145. Bliss index was calculated using the following equation: BI = $(E_A + E_B - E_A * E_B)/E_{AB}$. The Bliss index was calculated at each Olaparib concentration with constant NSC617145 concentration. The excess over Bliss (EOB) was calculated by subtracting the expected effect of the drug combination $(E_A + E_B - E_A * E_B)$ from the observed effect ($E_{AB}$) and multiplying that difference by 100. Here, $E_A$ and $E_B$ are the calculated effects of NSC617145 and Olaparib individually where the E = 1−% cell viability, and $E_{AB}$ is the actual effect of the two inhibitors in combination. More positive EOB values indicate stronger synergism[76].

**Statistics and reproducibility**. Statistical significance was determined by Student's two-tailed *t*-test, one-way or two-way ANOVA, and Mann–Whitney test using GraphPad Prism as indicated in individual figure legends. All experiments were performed at least twice unless otherwise specified. Sample size and *p* values are indicated in figure legends. Figures were prepared in Microsoft PowerPoint (Version 2108, Build 14326.20404).

**Reporting summary**. Further information on research design is available in the Nature Research Reporting Summary linked to this article.

## Data availability

All data of the present study are available within the article and its supplementary information files or available from the corresponding author upon request. Source data are provided with this paper.

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

## Acknowledgements

This work is supported by the National Institute on Aging and National Cancer Institute, NIH, and by R01 ES026184 to G.-L.M. We thank Dr. Lee Zou, Massachusetts General Hospital Cancer Center, Harvard Medical School for the kind gift of PEO1 and PEO4 cell lines. We thank Dr. Grant Stewart, Institute for Cancer and Genomic Sciences, University of Birmingham, Birmingham, United Kingdom for kindly providing the RECQL1-transfected HeLa/*RECQL1* KO cells that were generated by CRISPR KO in this study. We thank Dr. Jean-Yves Masson, Laval University Cancer Research Center, Québec City, Canada for kindly providing recombinant MRE11.

## Author contributions

A.D., K.B., J.A.S., H.T., and S.A. performed experiments. C.M.N., T.T. and G.-L.M. contributed the CRISPR WRN KO and WT cell lines. R.H.S. contributed 14C-NSC617145. S.K.S. and R.M.B. supervised xenograft and cell-based experiments, respectively. A.D. and R.M.B. wrote the manuscript draft. All authors provided critical comments and edited the manuscript.

## Funding

## Competing interests

The authors declare no competing interests.
