## [Peer Review File · Nature Communications]

REVIEWER COMMENTS

Reviewer #1 (Remarks to the Author):

BRCA2 is an essential factor to activate homologous recombination and repair double-strand breaks. BRCA2 is also involved in replication fork stabilization, as mutations in BRCA2 results in fork collapse and degradation. In this paper, the authors (Data et al.) discovered that WRN is required to limit uncontrolled fork degradation triggered by MRE11. The authors used DNA fiber assays to investigate the mechanism by which WRN stabilizes replication forks. When BRCA2 is depleted in WRN KO cells, defects in fork restart and fork degradation occur after replication stress. Fork restart defects is dependent only on WRN helicase activity, whereas fork degradation is dependent on both nuclease and helicase activity.

Furthermore, the authors use a potent inhibitor of WRN (NCS617145) which results in WRN trapping on DNA. This alleviated in the presence of the BRCA2 protein. Thus, in absence of BRCA2, the nascent stalled replication forks are degraded. The resulting double-strand DNA breaks are then repaired by NHEJ which favors genomic instability.

Even though the importance of the helicase WRN in the maintenance of cancer cell viability and fork replication protection is not new (Moles 2016 (in leukemia PMC5103433), Chan Nature 2019 (cancers with microsatellite instability PMC6580861)), the underlying mechanisms are not fully understood.

The paper is clearly written and contains well-documented data that support the conclusions. The data are interesting, novel, and relevant and substantiate the growing evidence for DNA helicase/nucleases in replication fork stability. The experiments are well-designed and the paper introduce new concepts such as WRN trapping or synthetic lethality with WRNi/PARPi in BRCA2 deficient cells. The paper also provide new therapeutic strategies to counteract PARP inhibitors resistance by using WRNi. Altogether, this paper is a strong contender for Nature communications pending the following revisions:

1. The data on fork stability is sound and clear. However, this is somewhat counterintuitive. Why WRN limits excessive fork degradation using helicase and nuclease activity ? Should this promote fork degradation in the absence of BRCA2 like other nucleases ? It is somewhat special that a nuclease protect the fork when BRCA2 is not there. I would welcome the authors to highlight this discrepancy in the discussion.
2. A follow-up question. In the WRN double mutant is DNA binding is affected so this could be a reason for the enhanced lack of protection in Fig 1d ?
3. In Fig. 1 the authors conclude that WRN limits excessive fork degradation upon BRCA2 loss using both helicase/nuclease functions of WRN. Later on (Fig 2a onwards), they focus mostly on the helicase mutant. The authors should use complementation assays with the nuclease mutant also to complement PEO1/PEO4 cell lines. As these cells might be difficult to transfect, I would suggest to use the AAVS1 integration system and select stable clones. This is quite an important point to validate their findings in different cell models.
4. Fig 1e: WRN^{-/-} cells seem to have less BRCA2 in the siCtrl than the WT, this could influence clonogenic efficiency. I wonder if this occurs systematically in WRN^{-/-} cells and what does this mean at the biological level ?
5. WRN trapping is likely the major source of DNA damage that results in killing activity. To prove this two (or more) cell lines (patient-derived) that express different levels of WRN could be used in killing curves. Can the level of expression of WRN be used to predict the effectiveness of the drug?
6. The authors have done some specificity controls (FANCI, RECQL1 binding assays on 14C-

radiolabeled NSC617145). Can the authors comment further on the specificity of the NSC617145 inhibitor? For instance in a pulldown/mass spectrometry assay? Also RECQL1 seems to bind to NSC617145 – is RECQL1 affecting some of the results presented? Although this might be technically challenging, one could overexpress RECQL1 in WRNi-treated cells and monitor fork stability/restart. If the inhibitor is targeting WRN only then they should see no effect.

7. Another control would be to add WRNi to WRN deficient cells and see whether the addition of WRNi is completely epistatic for: (i) survival in BRCA2 deficient cells and (ii) fork stability/restart. An additive effect would suggest that the inhibitor is targeting WRN but also something else.

8. The quantification of Figure 2d does not seem to match the results. In PEO-1 there is a constant increase of WRN bound to chromatin while the quantification at 2,4,8 hours shows constant levels of chromatin enrichment.

9. Fig 5 and b. DNA damage foci formation is very hard to see on the printout. The quality of the pictures needs to be improved. Zooming in, the DAPI is not clear, the nuclei are very pixelated (if printed the images are barely visible at all). The nuclei should be outlined and a larger image with less cells would be much better. Also, in Supp Figure 4, between a and d, only the staining changes but the nuclei look very different. In a they seem normal, however in d they have abnormal shapes. The authors should comment on this and amend their IF figures.

10. Also for the IFs, I see no mention of cell cycle markers which could be interesting since the study is focused on S-phase. I am wondering if the foci (especially p-DNA-PK foci) are related to fork stalling or general DNA damage processing.

11. Line 181. Fig 2a/b. In wt condition both WRN knockdown and WRNi reduce fork degradation. However, in the DLD cells WRNi results in stabilization of the fork. The authors should comment on this discrepancy.

12. Suppl. fig 2c: In the absence of WRNi and in the absence of BRCA2, WRN seems to have more affinity to chromatin. I am wondering if this is related to fork stalling or DNA damage processing.

13. Figure 6a/b and 7A. The authors should display synergy scores using Bliss. It is important to notice that the effect of WRNi on its own in PEO4 is much weaker than on the BRCA2-deficient PEO1 (these cells with only WRNi are already at 50% while the PEO4 remain at 90%). This would greatly influence the synergy scores. An actual mathematical analysis of these results would be helpful.

14. Model. At the biochemical level, what is the activity of WRN on reversed fork substrates? For instance, does WRN helicase activity remodel the substrate to allow nucleolytic degradation by MRE11? Does WRN binds to MRE11 to prevent fork degradation in vitro? It seems also unclear why WRN only binds reversed forks without activation of its intrinsic nuclease/helicase activities. The trapping model does not hold for all experiments presented (for instance, when WRN deficient cells are used). Further experiments would strengthen the model.

Minor comments

Fig 1b : Plasmid should replace cDNA

Fig 1e and Supp fig 6c - effcincy should read efficiency.

Representative images of the fork progression/protection experiments would be a good addition.

In the text the authors refer to the actual cell line rather than simply WT and BRCA2 mutant, they should stick to this labelling in the figure (for instance fig 2).

Fig 3e. For WRNi it says 8 uM instead of the micro symbol (please change the u for the mu) in both the treatment alone and with HU.

In Supp fig 5a, even though the top and bottom images are supposed to be the same cell line, they look very different. Looking just at the vehicle control, which should look rather similar, the shape, size and number of colonies looks fairly different from the top to the bottom. Please comment/explain.

Page 16, line 483, BRAC2 should read BRCA2.

Some blots are tilted (Fig. 1e, 2c, 3d, 3e) and this is not usual for the quality of figures of Nat Comm - please align properly.

Suppl fig 4. The b-actin blot does not show equal loading between samples.

Fig 3c. One can observe less OCR2 in the soluble fraction and chromatin fractions after WRN inhibition (4 uM). What does this mean in terms for general DNA replication ?

Fig 5c. Arrows should be red to make them more visible.

Methods

The methods section is general and the figure legends could provide more details. The authors give a range of number of cells that were plated and a range of days they were grown for, but they should specify which cell lines were plated in which amount and grown for a certain time. I would add more specific details for each cell line and/or experiment.

Reviewer #2 (Remarks to the Author):

In their work, Dadda and colleagues investigated on the role of WRN as regulator of fork processing in pathological conditions and analysed whether loss one of the two catalytic activities of this protein was preferentially involved in that process.

As prototypical model of fork degradation they chose BRCA2-deficient cells and found that loss of WRN and specifically of its helicase activity induces an even more elevated fork degradation phenotype with reduced replication fork progression during recovery.

They also demonstrated that degradation occurs through MRE11-dependent mechanism and involves channelling of processed forks through NHEJ with elevated instability at chromosomal level.

Most importantly, the authors showed that WRN inhibition by a small molecule trapping WRN in chromatin preferentially kills BRCA2-deficient cells and hypersensitize them to Olaparib.

Since WRN plays critical roles at different stages of the replication fork recovery process and has been repeatedly associated to regulation of fork degradation, these results help defining how WRN can act not only during normal replication but also under pathological conditions, which is important since mounting pieces of evidence highlight a critical role of WRN for the survival of specific types of tumors. Indeed, the most relevant authors' claim is that WRN helicase inhibitors might prove useful as anticancer therapies beyond the MSI tumours.

Collectively, the work is well conducted and scientifically sounding. The model is generally supported by the experimental data and results are clear. My major concern is that some critical controls are missing and that a little more effort is required to put the authors' finding, which are interesting and pretty novel, into the perspective of what is already known about the role of fork processing in normal and BRCA2-deficient cells and in view of what has been reported in the function of WRN at reversed

forks so far.

For instance, the effect of DNA2 inhibition should be investigated given that WRN and DNA2 cooperate in wild-type cells to perform a limited fork degradation at reversed forks. In addition, it would be useful to know if MUS81-dependent DSBs are formed downstream the hyper-degraded fork since DNA damage seems channelled to NHEJ and not BIR as reported to occur in BRCA2-deficient cells.

Moreover, it might be interesting to determine whether WRN inhibitions also exacerbates fork degradation in absence of Fanc proteins as the same lab demonstrated few years ago the existence of a similar synthetic sickness relationship between inhibition of WRN helicase and absence of Fanconi anemia.

The manuscript is well written, however, I would suggest to change the order of some figure to make the story more easy to follow and the main messages easy to catch.

Specific comments:

Figure 1. Experiments show that loss of WRN helicase reduces fork recovery. Fork recovery is evaluated as a ratio between labels but inset shows different classes of forks too. Basing on the graph, only the ratio has been considered. I suggest to remove inset or to show percentages of stalled vs restarting forks as well.

In BRCA2-deficient cells, both degradation and restart are affected by MRE11 inhibition. I know that Mirin data are shown later on in the manuscript but this is a good place to show them first and a good reason, at least to me, for reshuffling order of figures.

It would be interesting to include here a control showing what happens in siCTRL WRN-KO cells expressing different WRN mutants. Indeed, it has been previously reported that loss of WRN helicase reduces the partial fork degradation observed in a wild-type setting.

Figure 2. I would not say that WRN complementation rescues fork stability in BRCA2-depleted cells but rather that it rescues the hyper-degradation phenotype. It is known that there is still fork degradation in absence of BRCA2 but in the presence of WRN.

Figure 3. Panel D seems to do not be discussed in the text. Am I missing something?

In the panel D, it seems that WRN is more chromatin-bound when BRCA2 is mutated. This should be explicitly said as reinforce authors' hypothesis moreover it would be critical to demonstrate that accumulation of WRN occurs at fork.

DNA2 seems to get off chromatin when WRN helicase is inhibited. It is interesting and it is a point that deserves more data. Indeed, it has been reported that DNA2 is not implicated in fork degradation in BRCA2-deficient cells and that DNA2 depletion does not rescue the fork degradation phenotype in these cells.

It would be nice to assess whether DNA2 depletion or inhibition phenocopies WRN inhibition and if both inhibitors synergize. It might be possible that inhibition of WRN works by removing DNA2 and making forks more vulnerable to MRE11-EXO1 as the model proposed by the authors suggests. From this point of view, it would be interesting to assess whether MRE11 and/or EXO1 are more chromatin/fork recruited in BRCA2-depleted WRNi cells.

Figure 4. In the conclusion of the relevant paragraph, authors state that WRN inhibition engages BRCA2. In the model authors stated the opposite and chromatin data seems to support this claim, I would suggest to remove this sentence.

Fig. 4E. Given the cooperation between WRN and DNA2 in wild-type cells, I think that a control using DNA2 inhibitors or DNA2-depleted cells with or without WRNi should be shown.

Figure 5. While DNA damage data are solid, the authors should consider that DSBs arise very fast in BRCA2-deficient cells and that they seems to help fork restart through BIR. I would suggest to analyse the presence of DSBs at early time points too using some direct assay rather than (or in addition to) phosphorylated H2AX. This is an important point since hyper-degradation occurs in BRCA2-deficient cells also when MUS81 is depleted and no DSBs can be formed. I expect that DSBs are formed in

BRCA2-deficient/WRN inhibited cells also because MUS81 is hyperactivated in the absence of WRN, however, actual demonstration of their formation through a MUS81-dependent mechanism would rule out any possibility that MUS81 is inactive and would define whether a distinct pathway is engaged at the fork.

Authors show that NHEJ is engaged at the degraded forks. Is DNA-PK still active if degradation is inhibited using Mirin? How much of the additional chromosome instability conferred by WRNi in BRCA2-deficient cells correlates with NHEJ engagement and MRE11/EXO1-dependent degradation? Does this preferential NHEJ engagement occurs because of inability to activate RAD52-dependent BIR?

I would tone down the claim that WRNi is more potent than PARPi as it would require identification of IC50 etc to formal state this. Ditching this sentence would not harm the other solid conclusions of the paper.

Minor comments:

Given that authors acknowledge almost all factors contributing to fork stabilisation, I would also list RAD52 because it has unique mechanism of action and also a relevant role in BRCA2-deficient cells.

Reviewer #1

1. The data on fork stability is sound and clear. However, this is somewhat counterintuitive. Why WRN limits excessive fork degradation using helicase and nuclease activity? Should this promote fork degradation in the absence of BRCA2 like other nucleases? It is somewhat special that a nuclease protect the fork when BRCA2 is not there. I would welcome the authors to highlight this discrepancy in the discussion.

RESPONSE: We thank the Reviewer for posing these insightful questions. Our experimental results provide evidence that both helicase and exonuclease functions of WRN partially rescued nascent strand degradation under the BRCA2-depleted condition. Unlike BRCA2-proficient cells, WRN depletion in BRCA2-deficient cells did not rescue nascent DNA tract length, but instead further enhanced nascent strand degradation upon replication stalling, suggesting WRN prevents hyper-degradation of unprotected forks. In agreement with these results, we observed reduced nucleolytic cleavage of a synthetic reversed fork substrate by MRE11 nuclease in presence of WRN helicase. Thus, our results support a model in which stalled fork restoration by WRN helicase limits uncontrolled degradation of reversed forks by reduced availability of MRE11 degradation substrates³⁶ in BRCA2-deficient cells (see model shown in **Figure 7d** and also Discussion on page 19 of the revised manuscript).

We also found that WRN exonuclease partially rescues the hyper-degradation of unprotected forks in BRCA2-deficient cells. Previously, it was reported that in BRCA2-proficient cells WRN exonuclease activity suppresses nascent strand DNA degradation upon fork stalling induced by camptothecin²⁸. The importance of WRN exonuclease in fork protection was attributed to limitation of extensive nucleolytic activity by MRE11 and EXO1, thereby suppressing chromosomal instability²⁸. Thus, WRN exonuclease limits extensive fork degradation in both the wild-type and genetically deficient BRCA2 cells. Interestingly, CtIP which is an auxiliary factor of MRE11 in HR repair also protects forks during replication stress via its nuclease activity¹¹. These concepts have been addressed on page 19 of the Discussion.

2. A follow-up question. In the WRN double mutant is DNA binding is affected so this could be a reason for the enhanced lack of protection in Fig 1d?

RESPONSE: We performed this new experiment and the data are found in **Supplementary Figure 1e**. See Results section on page 6 and below:

“we found that both WRN-WT and WRN-K577M/E84A double mutant could bind to synthetic reversed fork DNA substrate with similar affinity (**Supplementary Figure 1e**), suggesting loss of both catalytic activities rather than loss of DNA binding prevented the WRN double mutant from rescuing hyper-degradation phenotype due to BRCA2 deficiency.”

3. In Fig. 1 the authors conclude that WRN limits excessive fork degradation upon BRCA2 loss using both helicase/nuclease functions of WRN. Later on (Fig 2a onwards), they focus mostly on the helicase mutant. The authors should use complementation assays with the nuclease mutant also to complement PEO1/PEO4 cell lines. As these cells might be difficult to transfect, I would suggest to use the AAVS1 integration system and select stable clones. This is quite an important point to validate their findings in different cell models.

RESPONSE: We attempted to create a WRN/BRCA2 double knockout in the PEO1 cells and also genetically engineered DLD1 BRCA2 knockout cells. However, we were unable to recover any double-deficient cells after multiple attempts, suggesting synthetic sickness or synthetic lethality in this genetic background. Using BRCA2-proficient DLD1 cells, we knocked out WRN. Using these DLD1/*WRN*^{-/-} cells

depleted of BRCA2, genetic complementation experiments were performed using plasmids encoding WRN-WT, WRN-K577M, WRN-E84A, and WRN-K577M/E84A. As shown in **Figure 1f** and **Supplementary Figure 1d**, WRN-WT, WRN-K577M or WRN-E84A, but not WRN-K577M/E84A, partially reduced the extensive fork degradation. These results are reported on page 6 of the revised manuscript.

4. Fig 1e: WRN^{-/-} cells seem to have less BRCA2 in the siCtrl than the WT, this could influence clonogenic efficiency. I wonder if this occurs systematically in WRN^{-/-} cells and what does this mean at the biological level ?

RESPONSE: See **Supplementary Figure 2c**. Text describing the result and possible implication is found in the Results section on page 7 and noted below:

“Of note, as judged by Western blot analysis of total cell lysate, BRCA2 level was reduced in *WRN*^{-/-} U2OS cells (**Supplementary Figure 2b**) and DLD1 cells (**Supplementary Figure 2c**), suggesting that phenotypes associated with WRN deficiency may reflect in part a loss of BRCA2; however, this requires further study.”

5. WRN trapping is likely the major source of DNA damage that results in killing activity. To prove this two (or more) cell lines (patient-derived) that express different levels of WRN could be used in killing curves. Can the level of expression of WRN be used to predict the effectiveness of the drug?

RESPONSE: We tested the WRNi sensitivity of LCLs previously characterized to have different levels of endogenous WRN (Moser et al. *Nucleic Acids Res.* 2000, PMID: 10606667). We include these data below for the reviewer’s inspection. Essentially, we found that the WRN^{+/+} cells are sensitive to the WRNi, whereas the WRN^{-/-} cells are resistant, consistent with our findings in the current study and published work (Aggarwal et al. *Cancer Res.* 2013, PMID: 23867477). WRN^{+/-} cells that display a >2-fold reduction in WRN protein compared to the WRN^{+/+} cells also displayed resistance to the WRNi. Previous studies have shown that cells from WRN heterozygotes display intermediate sensitivity to DNA damaging agents and are characterized by genomic instability (Ogburn et al., *Human Genet.* 1997 PMID: 9402954; Moser et al. *Nucleic Acids Res.* 2000, PMID: 10606667), suggesting a threshold loss of WRN displays WRNi sensitivity comparable to the WRN null cells.

6. The authors have done some specificity controls (FANCI, RECQL1 binding assays on 14C-radiolabeled NSC617145). Can the authors comment further on the specificity of the NSC617145 inhibitor? For instance in a pulldown/mass spectrometry assay? Also RECQL1 seems to bind to NSC617145 – is RECQL1 affecting some of the results presented? Although this might be technically challenging, one could overexpress RECQL1 in WRNi-treated cells and monitor fork stability/restart. If the inhibitor is targeting WRN only then they should see no effect.

RESPONSE: To address the specificity of the WRNi NSC617145, we performed several additional experiments.

First, we overexpressed RECQL1 in *BRCA2*-mutated PEO1 cells and determined that RECQL1 overexpression did not affect fork instability induced by the WRNi (**Supplementary Figure 3a**), suggesting the *in vivo* abundance of RECQL1 does not alter the fork-destabilizing effects of NSC617145. These results are presented in the Results section on page 8.

Secondly, we examined fork restart in HeLa RECQL1 knockout cells that had been stably transfected with an empty vector or wild-type RECQL1 plasmid. While WRN deficiency markedly attenuated the defect in WRNi-induced fork restart (**Supplementary Figure 4c**), we observed a significant reduction in fork restart efficiency in RECQL1^{-/-} cells upon WRNi treatment (**Supplementary Figure 4d**), indicating specificity of the WRNi to modulate fork restart. Consistent with a published role of RECQL1 in fork restart, we observed significantly reduced fork restart efficiency in RECQL1^{-/-} cells compared to isogenic RECQL1^{-/-} cells corrected by exogenous expression of RECQL1-WT (**Supplementary Figure 4d**). WRN helicase inhibition exasperated the fork restart defect in RECQL1^{-/-} cells (**Supplementary Figure 4d**), suggesting a specific effect of the WRNi that is independent of RECQL1 status. This information has been added to the Results section on page 9.

Thirdly, we performed experiments to assess the effect of the WRNi on viability of HeLa RECQL1 knockout cells transfected with empty vector or plasmid encoding RECQL1-WT. Our findings demonstrated that isogenic *RECQL1*-deficient and *RECQL1*-corrected cells exhibited similar sensitivity to

the WRNi (**Supplementary Figure 13e**), suggesting a RECQL1-independent effect. These results are presented on page 17 of the revised manuscript. Taken together, these results indicate that the cellular phenotypes induced by WRNi exposure are not mediated by modulation of RECQL1 activity. Consistent with this, in vitro experiments demonstrate only modest binding of purified recombinant RECQL1 to radiolabeled NSC617145 that is not statistically significant compared to the no protein control (**Figure 3a**), whereas the binding of WRN to ¹⁴C-NSC617145 was significant.

To further address the specificity of NSC617145, we have performed additional experiments with WRN-deficient cells, as described in the response to Reviewer Point #7 immediately below.

7. Another control would be to add WRNi to WRN deficient cells and see whether the addition of WRNi is completely epistatic for: (i) survival in BRCA2 deficient cells and (ii) fork stability/restart. An additive effect would suggest that the inhibitor is targeting WRN but also something else.

RESPONSE: We performed this experiment as suggested by the Reviewer and found that hypersensitivity to the WRNi upon BRCA2 depletion was also attenuated in BRCA2-depleted U2OS WRN^{-/-} cells as determined by WST-1 cytotoxicity assay (**Supplementary Figure 13f**). See also the response to point 3. Both WRN KO cells and BRCA2-depleted WRN KO cells are resistant to the WRNi compared to their respective WT cell lines (**Supplementary Figure 13f**). The new results are described on page 17 of the revised manuscript. In addition, fork stability was found to be significantly rescued upon WRN genetic depletion in BRCA2-mutated cells (PEO1) exposed to WRN inhibitor (**Figure 2a** of the revised manuscript). In new experiments, we assessed the specificity of the WRNi in fork restart assays using WT and WRN^{-/-} U2OS cells (**Supplementary Figure 4c**). WRN deficiency markedly attenuated the defect in WRNi-induced fork restart, indicating specificity of the WRNi to modulate fork restart. The new results are described on page 9.

8. The quantification of Figure 2d does not seem to match the results. In PEO-1 there is a constant increase of WRN bound to chromatin while the quantification at 2,4,8 hours shows constant levels of chromatin enrichment.

RESPONSE: We performed additional experiments and quantified the blots to address this issue, and determined that there was a dose-dependent enrichment of WRN in the chromatin fraction in both the WT and BRCA2-mutated cells (**Figure 3d** of revised manuscript).

9. Fig 5 and b. DNA damage foci formation is very hard to see on the printout. The quality of the pictures needs to be improved. Zooming in, the DAPI is not clear, the nuclei are very pixelated (if printed the images are barely visible at all). The nuclei should be outlined and a larger image with less cells would be much better. Also, in Supp Figure 4, between a and d, only the staining changes but the nuclei look very different. In a they seem normal, however in d they have abnormal shapes. The authors should comment on this and amend their IF figures.

RESPONSE: Following the reviewer's advice, we have replaced the previous figures with ones showing a reduced number of cells but of higher resolution and the nuclei have been outlined (**Supplementary Figure 7a** and **Supplementary Figure 8a** in the revised manuscript). With regard to the comment about γ H2AX and 53BP1 staining and difference in appearance of nuclei in the original **Supplementary Figure 4a, 4d**, the experiments are separate because the primary antibodies against γ H2AX and 53BP1 were raised in the same species (mouse). We have amended the figures to provide ones of higher resolution and outlined nuclei. See **Supplementary Figures 7b, 7d** in the revised manuscript.

10. Also for the IFs, I see no mention of cell cycle markers which could be interesting since the study is

focused on S-phase. I am wondering if the foci (especially p-DNA-PK foci) are related to fork stalling or general DNA damage processing.

RESPONSE: We thank the Reviewer for this comment and have addressed it by performing new experiments. We have now included new data from experiments in which the cells were co-stained with the S-phase specific marker PCNA. Because the primary PCNA antibody is raised in mouse, we used a DNA-PKcs pS2056 and γ H2AX antibodies raised in rabbit for immunofluorescence experiments. The new results are described below and can be found in **Figure 4a** and **Figure 5a**, as well as on page 13 and 14, respectively in the Results section of the revised manuscript.

“To determine if DSBs induced by exposure to the WRNi occur preferentially in replicating cells, we co-stained for γ H2AX and PCNA (**Figure 4a**). γ H2AX was predominantly observed in PCNA-positive cells, suggesting replication-associated DNA damage (**Figure 4a**). To determine if the aberrant NHEJ activation occurs in replicating cells, we monitored DNA-PKcs phosphorylation at serine 2056 (DNA-PKcs pS2056)⁵¹ in cells co-stained for PCNA (**Figure 5a**). We observed DNA-PKcs pS2056 formation primarily in PCNA-positive cells, suggesting WRNi-induced NHEJ activation in cells actively performing DNA synthesis. Interestingly, DNA-PKcs pS2056 foci were reduced in *BRCA2*-mutated cells exposed to Mirin (**Figure 5a**), suggesting induction of NHEJ is related to MRE11-mediated fork degradation.”

11. Line 181. Fig 2a/b. In wt condition both WRN knockdown and WRNi reduce fork degradation. However, in the DLD cells WRNi results in stabilization of the fork. The authors should comment on this discrepancy.

RESPONSE: The WRNi has a greater effect on fork stalling in DLD1 cells (**Figure 2e**) compared to PEO4 cells (**Supplementary Figure 5c**). This difference may help to explain why we observed a small decrease in IdU/CldU ratio in DLD1 WT cells (**Supplementary Figure 3b**) compared to a slight increase in PEO4 cells (**Figure 1h**). In addition, the observed discrepancy between PEO4 (Ovarian cancer cells) and DLD1 (Colorectal cancer cells) may reflect differences in lineage.

12. Suppl. fig 2c: In the absence of WRNi and in the absence of BRCA2, WRN seems to have more affinity to chromatin. I am wondering if this is related to fork stalling or DNA damage processing.

RESPONSE: We appreciate the comment made by the Reviewer and have addressed it with new experiments. The new results have been added (see below) and are found on page 12 of the revised manuscript.

“Furthermore, even in the absence of NSC617145, WRN showed enrichment in the chromatin fraction of *BRCA2*-mutated cells (**Figure 3d**). To test if this was related to fork stalling or general DNA damage processing, we examined WRN immunofluorescence intensity in PEO1 and PEO4 replicating cells counterstained with the S-phase-specific marker PCNA (**Supplementary Figure 6c**). As reported previously⁴⁴, we observed that WRN primarily localized to the nucleoli under unperturbed condition. However, compare to WT cells, PCNA-positive *BRCA2*-mutated cells showed increased WRN intensity, suggesting WRN accumulation at forks in the absence of BRCA2. Of note, consistent with published data that actively proliferating cancer cell lines display elevated level of WRN⁴⁵, we detected substantially enhanced WRN immunofluorescence in PCNA-positive cells as compared to PCNA-negative cells (**Supplementary Figure 6c**).”

13. Figure 6a/b and 7A. The authors should display synergy scores using Bliss. It is important to notice that the effect of WRNi on its own in PEO4 is much weaker than on the BRCA2-deficient PEO1 (these cells with only WRNi are already at 50% while the PEO4 remain at 90%). This would greatly influence the

synergy scores. An actual mathematical analysis of these results would be helpful.

RESPONSE: We have performed the mathematical analysis by determining Coefficient of Drug interaction (CDI) and Bliss analysis. Both analyses indicate synergism of WRNi and Olaparib in PEO1 and PEO4 cells. This has been added to the manuscript on page 18 and written below:

“We further assessed drug-drug interaction effects by analyzing coefficient of drug interaction (CDI) and observed synergistic cell killing ($CDI < 1$) by the WRNi-olaparib combination in both PEO1 and PEO4 cells (**Figure 7b**). However, at the lowest concentration of olaparib (0.05 μ M), there was much stronger synergism ($CDI < 0.7$) in PEO1 cells compared to PEO4 cells (**Figure 7b**). Bliss synergy analysis further revealed synergistic cell killing by WRNi and olaparib as indicated by positive Excess over Bliss (EOB > 0) synergy scores (**Figure 7c**).”

14. Model. At the biochemical level, what is the activity of WRN on reversed fork substrates ? For instance, does WRN helicase activity remodel the substrate to allow nucleolytic degradation by MRE11 ? Does WRN binds to MRE11 to prevent fork degradation in vitro ? It seems also unclear why WRN only binds reversed forks without activation of its intrinsic nuclease/helicase activities. The trapping model does not hold for all experiments presented (for instance, when WRN deficient cells are used). Further experiments would strengthen the model.

RESPONSE: We have performed new biochemical experiments that assessed WRN’s catalytic activity on an oligonucleotide-based reversed fork DNA substrate. The results are found in **Figure 1c, Figure 1d**, described on page 5 of the revised manuscript and written below:

“In agreement with these results, purified recombinant WRN-WT or WRN-E84A catalyzed fork restoration from a model reversed fork (“chicken-foot”) DNA substrate *in vitro* whereas WRN-K577M failed to do so (**Figure 1c and 1d**), suggesting fork restoration is supported by WRN ATPase/helicase activity. These results suggest that restoration of intact replications forks by WRN helicase is critical for efficient replication resumption following fork stalling when BRCA2 function is lost.”

We also tested if WRN inhibits MRE11 nuclease on the model reversed for substrate. Because WRN exonuclease has the same polarity as MRE11 (3’ to 5’), we used a purified recombinant WRN exonuclease-dead protein (WRN-E84A) in reaction mixtures. As shown in **Supplementary Figure 1b** of the revised manuscript, WRN suppressed MRE11 exonuclease activity on the regressed fork. The results are described on page 5 in the revised manuscript.

We have revised the model (**Figure 7d** of the revised manuscript) to depict WRN’s fork restoration activity that is supported by the biochemical data. The model in **Figure 7e** depicts WRN trapping on the DNA caused by the WRNi. To investigate this further, we examined MRE11 chromatin enrichment in cells exposed to the WRNi. We observed a strong dose-dependent enrichment of MRE11 in the chromatin fraction (**Figure 2c**), supporting the model that WRNi causes rapid accumulation of replication fork intermediates suitable for MRE11 engagement and subsequent degradation of nascent DNA if not protected by the fork-stabilizing protein BRCA2. This information has been added to the revised manuscript on page 10.

Minor comments

Fig 1b : Plasmid should replace cDNA

RESPONSE: Corrected.

Fig 1e and Supp fig 6c - effcincy should read efficiency.

RESPONSE: Corrected.

Representative images of the fork progression/protection experiments would be a good addition.

RESPONSE: Representative images of the fiber experiments underlying key results are now added to the revised manuscript. Please see **Supplementary Figure 1a, 1c, 1d, 2d, 2e, 4a, 4b** and **5b**.

In the text the authors refer to the actual cell line rather than simply WT and BRCA2 mutant, they should stick to this labelling in the figure (for instance fig 2).

RESPONSE: Corrected.

Fig 3e. For WRNi it says 8 uM instead of the micro symbol (please change the u for the mu) in both the treatment alone and with HU.

RESPONSE: Corrected.

In Supp fig 5a, even though the top and bottom images are supposed to be the same cell line, they look very different. Looking just at the vehicle control, which should look rather similar, the shape, size and number of colonies looks fairly different from the top to the bottom. Please comment/explain.

RESPONSE: We have addressed this issue by carefully repeating the experiments done at the same time using the DLD1/WT and DLD1/BRCA2^{-/-} cells exposed to the WRNi or olaparib. The previous figure is replaced by **Supplementary Figure 9a** and the quantitation is shown in **Figure 6a**.

Page 16, line 483, BRAC2 should read BRCA2.

RESPONSE: Corrected.

Some blots are tilted (Fig. 1e, 2c, 3d, 3e) and this is not usual for the quality of figures of Nat Comm - please align properly.

RESPONSE: We have carefully examined all figures and adjusted them appropriately.

Suppl fig 4. The b-actin blot does not show equal loading between samples.

RESPONSE: We repeated the experiments to address this. Please see **Supplementary Figure 7c** showing equal loading of b-actin.

Fig 3c. One can observe less OCR2 in the soluble fraction and chromatin fractions after WRN inhibition (4 uM). What does this mean in terms for general DNA replication ?

RESPONSE: We performed additional experiments to carefully address this issue. The ORC protein level was quantified from three independent experiments. These data did not show any significant difference in ORC2 level as a function of WRNi dose for either the soluble or chromatin fractions (see below). We have replaced the previous immunoblot with a new one that is more representative (**Figure 3c**).

One-way ANOVA with Dunnett's multiple comparison test

Fig 5c. Arrows should be red to make them more visible.

RESPONSE: Corrected. Please see revised **Figure 5b**.

The methods section is general and the figure legends could provide more details. The authors give a range of number of cells that were plated and a range of days they were grown for, but they should specify which cell lines were plated in which amount and grown for a certain time. I would add more specific details for each cell line and/or experiment.

RESPONSE: We have carefully addressed this issue by providing more detailed information in both the Methods section and figure legends. See revised manuscript. Note that for fork stability assay shown in **Figure 1h** of the revised manuscript, 4 μM WRNi was used rather than 2 μM, which was a labeling mistake in the originally submitted manuscript (**Figure 2b** of original manuscript).

Reviewer #2

Collectively, the work is well conducted and scientifically sounding. The model is generally supported by the experimental data and results are clear. My major concern is that some critical controls are missing and that a little more effort is required to put the authors' finding, which are interesting and pretty novel, into the perspective of what is already known about the role of fork processing in normal and BRCA2-deficient cells and in view of what has been reported in the function of WRN at reversed forks so far.

For instance, the effect of DNA2 inhibition should be investigated given that WRN and DNA2 cooperate in wild-type cells to perform a limited fork degradation at reversed forks.

RESPONSE: To address the potential role of DNA2 in fork stability when BRCA2 is deficient, we performed new DNA fiber experiments and added these results to the revised manuscript

(**Supplementary Figure 3c; Figure 2b**). The new data are described in the Results section on page 8 and 10, and found below:

“Because DNA2 nuclease operates in conjunction with WRN helicase to allow limited fork processing in BRCA2 WT cells²⁶, we tested the stability of HU-stalled forks upon DNA2 inhibition in BRCA2-mutated cells using a previously characterized small molecule inhibitor of DNA2 nuclease⁴⁰ (**Supplementary Figure 3c**). Unlike WRN helicase inhibition, DNA2 inhibition did not exacerbate fork degradation in BRCA2-mutated PEO1 cells; moreover, the DNA2i did not synergize with the WRNi in its effect on fork stability (**Supplementary Figure 3c**).”

“Unlike the case for MRE11, pre-exposure to a small molecule inhibitor of DNA2 nuclease³⁹ did not rescue WRNi-induced fork degradation in PEO1 cells (**Figure 2b**), suggesting that MRE11 plays a more dominant role in nucleolytic degradation of stalled forks in WRNi-treated *BRCA2*-mutated cells.”

In addition, it would be useful to know if MUS81-dependent DSBs are formed downstream the hyper-degraded fork since DNA damage seems channelled to NHEJ and not BIR as reported to occur in BRCA2-deficient cells.

RESPONSE: To address the role of MUS81 in DSB formation, we performed neutral comet assays using BRCA2-mutated cells depleted (or not) of MUS81. The new data can be found in **Figure 4b** and **Supplementary Figure 7e** of the revised manuscript and are described in the Results section on page 14. See below:

“Previously, it was shown that WRN loss of function results in replication fork collapse and subsequent formation of MUS81-mediated DSBs at collapsed forks²³. Moreover, MUS81-dependent DSBs are generated following hyper-degradation of stalled forks in BRCA2-deficient cells³⁶. Therefore, we assayed for DSB formation upon WRN helicase inhibition by neutral Comet assays (**Figure 4b**). Compared to WT PEO4 cells, *BRCA2*-mutated PEO1 cells displayed elevated DSBs upon WRNi treatment (**Figure 4b**). However, increased DSB formation in *BRCA2*-mutated cells was found to be significantly abrogated upon MUS81 depletion (**Figure 4b** and **Supplementary Figure 7e**). These observations suggest that WRN helicase inhibition results in MUS81-dependent DSBs as a consequence of stalled fork hyper-degradation in *BRCA2*-mutated cells.”

Moreover, it might be interesting to determine whether WRN inhibitions also exacerbates fork degradation in absence of Fanc proteins as the same lab demonstrated few years ago the existence of a similar synthetic sickness relationship between inhibition of WRN helicase and absence of Fanconi anemia.

RESPONSE: To determine if WRN helicase inhibition exasperates fork instability in cells defective in the FA pathway, we performed the corresponding DNA fiber experiments using a FANCD2-deficient cell line. The results from these experiments demonstrated that unlike BRCA2-deficient cells, the fork stability of FANCD2-deficient cells is not compromised by the WRNi, suggesting that WRN helicase has distinct roles to maintain forks when either BRCA2 or FANCD2 is not present. These data are found in **Supplementary Figure 3d** of the revised manuscript and described on page 8 of the Results section. See below:

“WRN helicase inhibition was previously reported to cause mitomycin c hypersensitivity of cells lacking FANCD2³⁸, a protein important for fork stability and implicated in the Fanconi anemia pathway of interstrand crosslink repair⁸. Therefore, we tested whether WRN helicase inhibition triggers elevated fork instability in FANCD2-deficient cells. Unlike BRCA2-deficient cells, WRN helicase inhibition did not further

exacerbate fork degradation in *FANCD2*^{-/-} PD20 cells (**Supplementary Figure 3d**), suggesting that the role of WRN helicase activity in fork maintenance is distinct when BRCA2 or FANCD2 is absent.”

The manuscript is well written, however, I would suggest to change the order of some figure to make the story more easy to follow and the main messages easy to catch.

RESPONSE: We thank the Reviewer for this input to restructure the manuscript to enhance the logical flow of the study and communication of experimental results. Below are listed the key changes made to improve the flow of the manuscript:

The fork stability and fork restart assays in *BRCA2* WT and mutated cells depleted of WRN or exposed to the WRNi have been moved from **Figure 2** of the original manuscript and combined with **Figure 1**.

Figure 4 of the original manuscript detailing the results from experiments assessing the impact of the WRNi on *BRCA2*-mutated and wild-type cells that have been depleted of the DNA fork remodeling enzymes has been moved to **Figure 2** of the revised manuscript. **Figure 2** describes hyper-degradation of stalled forks in *BRCA2*-deficient cells as a function of WRN helicase inhibition.

Experimental results of DNA fiber assays showing the effect of Mirin-mediated MRE11 nuclease inhibition on fork stability in WRNi-treated *BRCA2*-mutated cells, as well as the new data showing i) chromatin enrichment of MRE11 upon WRNi treatment, and ii) effect of DNA2 inhibition on fork degradation induced by WRNi exposure, have been presented in **Figure 2** of the revised manuscript. This move should satisfy a comment raised by the Reviewer below as well.

Specific comments:

Figure 1. Experiments show that loss of WRN helicase reduces fork recovery. Fork recovery is evaluated as a ratio between labels but inset shows different classes of forks too. Basing on the graph, only the ratio has been considered. I suggest to remove inset or to show percentages of stalled vs restarting forks as well.

RESPONSE: The inset (in **Figure 1b** of the revised manuscript) has been removed.

In *BRCA2*-deficient cells, both degradation and restart are affected by MRE11 inhibition. I know that Mirin data are shown later on in the manuscript but this is a good place to show them first and a good reason, at least to me, for reshuffling order of figures.

RESPONSE: The content of **Figure 4** in the original manuscript has been shifted to **Figure 2** of the revised manuscript so that the Mirin data appear earlier in the manuscript, as suggested by the Reviewer. Accordingly, we have changed the Results section in the order of results presentation to reflect this alteration.

It would be interesting to include here a control showing what happens in siCTRL WRN-KO cells expressing different WRN mutants. Indeed, it has been previously reported that loss of WRN helicase reduces the partial fork degradation observed in a wild-type setting.

RESPONSE: We have performed the requested experiment and the new data added to the manuscript are found in **Supplementary Figure 2a**. The new results are discussed on page 6 and found below:

“Of note, we observed partial fork degradation upon complementation with either WRN-WT or WRN-E84A in control siRNA transfected U2OS/*WRN*^{-/-} cells (**Supplementary Figure 2a**), suggesting limited nucleolytic processing of the stalled forks by WRN helicase when BRCA2 is present, as reported previously²⁶.”

Figure 2. I would not say that WRN complementation rescues fork stability in BRCA2-depleted cells but rather that it rescues the hyper-degradation phenotype. It is known that there is still fork degradation in absence of BRCA2 but in the presence of WRN.

RESPONSE: We thank the Reviewer for pointing this out and have changed the language on page 6. See below:

“Complementation with WT WRN partially rescued the hyper-degradation phenotype caused by BRCA2 depletion in U2OS/*WRN*^{-/-} cells (**Figure 1e**)”

Figure 3. Panel D seems to do not be discussed in the text. Am I missing something?

In the panel D, it seems that WRN is more chromatin-bound when BRCA2 is mutated. This should be explicitly said as reinforce authors' hypothesis moreover it would be critical to demonstrate that accumulation of WRN occurs at fork.

RESPONSE: We thank the Reviewer for pointing this out. We have added a statement to the Results section on page 12 to indicate the chromatin enrichment of WRN in BRCA2-mutated cells treated with the WRNi.

“Compared to WT cells, NSC617145-induced chromatin enrichment of WRN was elevated in *BRCA2*-mutated cells, suggesting that WRN accumulates at forks when BRCA2 is absent (**Figure 3d**).”

DNA2 seems to get off chromatin when WRN helicase is inhibited. It is interesting and it is a point that deserves more data. Indeed, it has been reported that DNA2 is not implicated in fork degradation in BRCA2-deficient cells and that DNA2 depletion does not rescue the fork degradation phenotype in these cells.

RESPONSE: In the original manuscript submission, the experimental data suggested that the fraction of DNA2 in chromatin was not enriched, as we observed for WRN. To examine this further, we have repeated the experiments and quantified the amount of DNA2 in the chromatin fraction obtained from PEO4 and PEO1 cells that had been exposed to the WRNi. These new data are found in **Supplementary Figure 6d**, which now replace the DNA2 immunoblot in the original manuscript. Quantitation of DNA2, shown in **Supplementary Figure 6d** demonstrate that there is no significant reduction in chromatin-bound DNA2 in both cell types irrespective of WRNi exposure. This is stated on page 13 of the Results section.

“Importantly, neither DNA topoisomerase I (**Figure 3f**) nor DNA2 nuclease (**Supplementary Figure 6d**) was found to accumulate in the chromatin fraction in NSC617145-treated cells, suggesting a specific effect of the compound to bind and trap WRN on genomic DNA.”

It would be nice to assess whether DNA2 depletion or inhibition phenocopies WRN inhibition and if both inhibitors synergize. It might be possible that inhibition of WRN works by removing DNA2 and making forks more vulnerable to MRE11-EXO1 as the model proposed by the authors suggests.

RESPONSE: We have performed a new set of experiments to address this point. The new results are shown in **Supplementary Figure 3c** and described in the Results section on page 8, as detailed below:

“Because DNA2 nuclease operates in conjunction with WRN helicase to allow limited fork processing in BRCA2 WT cells²⁶, we tested the stability of HU-stalled forks upon DNA2 inhibition in BRCA2-mutated cells using a previously characterized small molecule inhibitor of DNA2 nuclease⁴⁰ (**Supplementary Figure 3c**). Unlike WRN helicase inhibition, DNA2 inhibition did not exacerbate fork degradation in BRCA2-mutated PEO1 cells; moreover, the DNA2i did not synergize with the WRNi in its effect on fork stability (**Supplementary Figure 3c**).”

From this point of view, it would be interesting to assess whether MRE11 and/or EXO1 are more chromatin/fork recruited in BRCA2-depleted WRNi cells.

RESPONSE: We have performed new experiments to address the recruitment of MRE11 and EXO1 to chromatin in BRCA2-deficient cells exposed to the WRNi. The new data can be found in **Figure 2c** and in the Results section on page 10. They are also found below:

“Consistent with this notion, MRE11 was preferentially enriched in the chromatin fraction obtained from PEO1 cells exposed to WRNi compared to PEO4 cells (**Figure 2c**). In BRCA2-deficient cells, degradation of HU-stalled fork initiated by MRE11 is further extended by EXO1 nuclease³⁶. However, we did not observe chromatin enrichment of EXO1 following WRNi treatment (**Figure 2c**). MRE11 and EXO1 may independently act on stalled forks in BRCA-deficient cells, as combined loss of the two nucleases has been reported to further rescue fork stability in these cells³⁶. Our results suggest that WRNi causes rapid accumulation of replication fork intermediates suitable for MRE11 engagement and subsequent degradation of nascent DNA if not protected by the fork-stabilizing protein BRCA2.”

Figure 4. In the conclusion of the relevant paragraph, authors state that WRN inhibition engages BRCA2. In the model authors stated the opposite and chromatin data seems to support this claim, I would suggest to remove this sentence.

RESPONSE: As requested, we have removed the sentence.

Fig. 4E. Given the cooperation between WRN and DNA2 in wild-type cells, I think that a control using DNA2 inhibitors or DNA2-depleted cells with or without WRNi should be shown.

RESPONSE: We have performed the requested experiment and determined that the presence of the DNA2 nuclease inhibitor did not rescue fork degradation induced by WRNi. The results are shown in **Figure 2b**, described in the Results section on page 10, and also detailed below:

“Unlike the case for MRE11, pre-exposure to a small molecule inhibitor of DNA2 nuclease³⁹ did not rescue WRNi-induced fork degradation in PEO1 cells (**Figure 2b**), suggesting that MRE11 plays a more dominant role in nucleolytic degradation of stalled forks in WRNi-treated BRCA2-mutated cells.”

In **Supplementary Figure 3c**, the data from DNA fiber experiments using PEO4 and PEO1 cells exposed to the DNA2i (in the absence of WRNi) are shown, suggesting DNA2 inhibition did not exacerbate fork

degradation in *BRCA2*-mutated or wild-type cells. This finding is stated on page 8 of the Results section and is consistent with a published study (Lemacon et al., Nat Comm (2017); PMID: 29038425), suggesting that DNA2 does not contribute to fork degradation in *BRCA2*-deficient cells.

Figure 5. While DNA damage data are solid, the authors should consider that DSBs arise very fast in *BRCA2*-deficient cells and that they seem to help fork restart through BIR. I would suggest to analyse the presence of DSBs at early time points too using some direct assay rather than (or in addition to) phosphorylated H2AX. This is an important point since hyper-degradation occurs in *BRCA2*-deficient cells also when MUS81 is depleted and no DSBs can be formed.

RESPONSE: To address the Reviewer's point, we have examined γ H2AX at the 6-hr time point and performed neutral Comet analysis after a 6 hr WRNi exposure. The results from γ H2AX immunofluorescence are shown in **Figure 4a** and Comet analysis in **Figure 4b**. Results are presented on page 13 and 14 of the manuscript and also below:

"To determine if DSBs induced by exposure to the WRNi occur preferentially in replicating cells, we co-stained for γ H2AX and PCNA (**Figure 4a**). γ H2AX was predominantly observed in PCNA-positive cells, suggesting replication-associated DNA damage (**Figure 4b**)."

"Previously, it was shown that WRN loss of function results in replication fork collapse and subsequent formation of MUS81-mediated DSBs at collapsed forks²³. Moreover, MUS81-dependent DSBs are generated following hyper-degradation of stalled forks in *BRCA2*-deficient cells³⁶. Therefore, we assayed for DSB formation upon WRN helicase inhibition by neutral Comet assays (**Figure 4b**). Compared to WT PEO4 cells, *BRCA2*-mutated PEO1 cells displayed elevated DSBs upon WRNi treatment (**Figure 4b**). However, increased DSB formation in *BRCA2*-mutated cells was found to be significantly abrogated upon MUS81 depletion (**Figure 4b**). These observations suggest that WRN helicase inhibition results in MUS81-dependent DSBs as a consequence of stalled fork hyper-degradation in *BRCA2*-mutated cells."

I expect that DSBs are formed in *BRCA2*-deficient/WRN inhibited cells also because MUS81 is hyperactivated in the absence of WRN, however, actual demonstration of their formation through a MUS81-dependent mechanism would rule out any possibility that MUS81 is inactive and would define whether a distinct pathway is engaged at the fork.

RESPONSE: As mentioned in the response to the previous comment, we performed experiments with MUS81-depleted cells and analyzed for DSBs by neutral Comet assays. See **Figure 4b**.

"Therefore, we assayed for DSB formation upon WRN helicase inhibition by neutral Comet assays (**Figure 4b**). Compared to WT PEO4 cells, *BRCA2*-mutated PEO1 cells displayed elevated DSBs upon WRNi treatment (**Figure 4b**). However, increased DSB formation in *BRCA2*-mutated cells was found to be significantly abrogated upon MUS81 depletion (**Figure 4b** and **Supplementary Figure 7e**)."

Authors show that NHEJ is engaged at the degraded forks. Is DNA-PK still active if degradation is inhibited using Mirin?

RESPONSE: We performed new experiments and determined that DNA-PKcs pS2056 foci were reduced in *BRCA2*-mutated cells exposed to Mirin (**Figure 5a**), suggesting induction of NHEJ is related to MRE11-mediated fork degradation. The data can be found in **Figure 5a** and described in Results section on page 14, as well as related below:

“To determine if the aberrant NHEJ activation occurs in replicating cells, we monitored DNA-PKcs phosphorylation at serine 2056 (DNA-PKcs pS2056)⁵¹ in cells co-stained for PCNA (**Figure 5a**). We observed DNA-PKcs pS2056 formation primarily in PCNA-positive cells, suggesting WRNi-induced NHEJ activation in cells actively performing DNA synthesis. Interestingly, DNA-PKcs pS2056 foci were reduced in *BRCA2*-mutated cells exposed to Mirin (**Figure 5a**), suggesting induction of NHEJ is related to MRE11-mediated fork degradation. In *BRCA2*-deficient cells, hyper-degraded forks are rescued by a break-induced replication (BIR) repair mechanism following MUS81-mediated cleavage³⁴.”

How much of the additional chromosome instability conferred by WRNi in *BRCA2*-deficient cells correlates with NHEJ engagement and MRE11/EXO1-dependent degradation?

RESPONSE: We addressed this point with new experiments found in **Figure 5c** and described in the Results section on page 15, as well as below:

“Inhibition of MRE11 nuclease alleviated gross chromosomal aberrations in *BRCA2*-mutated PEO1 cells (**Figure 5c**), suggesting that WRNi-induced genomic instability was primarily attributed to MRE11-mediated fork degradation. We also observed a significant reduction in chromosomal abnormalities upon DNA-PKcs inhibition (**Figure 5c**), suggesting NHEJ induction contributes to the heightened genomic instability in WRNi-treated *BRCA2*-mutated cells. Collectively, these observations suggest that blocking WRN helicase function pharmacologically elicits DNA damage and error-prone NHEJ, leading to genomic instability in cells lacking functional *BRCA2*.”

Does this preferential NHEJ engagement occur because of inability to activate RAD52-dependent BIR?

RESPONSE: We performed additional experiments and determined that RAD52 depletion in WRNi-treated PEO1 cells causes further increased percentage of stalled forks. The results are found in **Supplementary Figure 8b**, described on page 14 and 15 of the Results section, and below:

“Rad52, a key mediator of HR has been implicated in repair of collapsed replication fork⁵² and recovery of stalled forks¹⁶. We, therefore analyzed percentage of stalled forks in WRNi-treated PEO1 cells upon knockdown of RAD52 (**Supplementary Figure 8b**). Either WRN helicase inhibition or RAD52 depletion resulted in significantly elevated fork stalling. RAD52 knockdown in WRNi-treated PEO1 cells further increased the percentage of stalled forks, suggesting a role of RAD52-dependent BIR to aid in the recovery of hyper-degraded forks. However, given the observed increase in NHEJ (**Supplementary Figure 8a** and **Figure 5a**), our results collectively suggest that in *BRCA2*-mutated cells, a subset of collapsed forks downstream of WRNi-induced hyper-degradation also engage NHEJ for repair”.

I would tone down the claim that WRNi is more potent than PARPi as it would require identification of IC50 etc to formal state this. Ditching this sentence would not harm the other solid conclusions of the paper.

RESPONSE: We have removed the sentence.

Minor comments:

Given that authors acknowledge almost all factors contributing to fork stabilisation, I would also list RAD52 because it has unique mechanism of action and also a relevant role in *BRCA2*-deficient cells.

RESPONSE: RAD52 has now been added to the list of replication stress response proteins important for fork protection in the Introduction on page 3.

REVIEWERS' COMMENTS

Reviewer #1 (Remarks to the Author):

My concerns have been addressed satisfactorily and the paper has been greatly improved. In particular, the final model is more accurate I believe. I congratulate the authors for this nice study.

Reviewer #2 (Remarks to the Author):

I have read the revised version of the manuscript and the point-by-point replies from the authors.

Taking into account the comments raised by the other reviewer and mine, I think that the authors have done a great job to improve the manuscripts and effectively deal with my comments and also those from the other reviewer.

I think that the manuscript is suitable for publication but have a couple of minor comments for the authors with few points unclear to me in the discussion.

Indeed:

1. I find incomplete to state that WRN is required for the recovery of stalled forks in the absence of BRCA2 without taking into account that previous data indicate that recovery occurs downstream MUS81-dependent cleavage (MUS81-depletion reduces recovery in BRCA2-deficient cells).

I'm not asking for additional experiments but even some speculation of what may happen might be useful to avoid confusion to readers. May be, although not formally tested, WRN and RAD52-MUS81-PolD1 are two axes used by cells to restart forks in a BRCA2-deficient background?

2. Statement about accumulation of reversed forks in WRNi-treated BRCA2-deficient cells is incomplete too without taking into account that in the absence of BRCA2 most reversed forks are already degraded and transformed into DSBs. As stated, one might wonder whether WRNi counteracts MUS81-dependent cleavage, which is not the case as shown by the authors.

Thus, I would strongly advise authors to revise the final paragraph of the discussion to reconcile their own data with data from previous works and the first scheme of the model to include previous data from other groups as well.

REVIEWERS' COMMENTS

Reviewer #1 (Remarks to the Author):

My concerns have been addressed satisfactorily and the paper has been greatly improved. In particular, the final model is more accurate I believe. I congratulate the authors for this nice study.

RESPONSE: We are pleased that the Reviewer is satisfied with the manuscript.

Reviewer #2 (Remarks to the Author):

I have read the revised version of the manuscript and the point-by-point replies from the authors.

Taking into account the comments raised by the other reviewer and mine, I think that the authors have done a great job to improve the manuscripts and effectively deal with my comments and also those from the other reviewer.

RESPONSE: We are happy to receive this positive news from the Reviewer.

I think that the manuscript is suitable for publication but have a couple of minor comments for the authors with few points unclear to me in the discussion.

Indeed:

1. I find incomplete to state that WRN is required for the recovery of stalled forks in the absence of BRCA2 without taking into account that previous data indicate that recovery occurs downstream MUS81-dependent cleavage (MUS81-depletion reduces recovery in BRCA2-deficient cells).

I'm not asking for additional experiments but even some speculation of what may happen might be useful to avoid confusion to readers. May be, although not formally tested, WRN and RAD52-MUS81-PolD1 are two axes used by cells to restart forks in a BRCA2-deficient background?

RESPONSE: We have modified the corresponding text in the Discussion to accommodate the Reviewer's comment. The revised text found on page 20, lines 580-585 are as follows: "Rad52-dependent BIR promotes stalled fork recovery under conditions of replication stress¹⁶. Moreover, in *BRCA2*-deficient cells, resected reversed forks are rescued by a POLD3-dependent BIR mechanism following MUS81-mediated fork cleavage³⁵. Thus, although not formally tested, it is

plausible that WRN and RAD52-MUS81-POLD3 are two axes used by cells to restart forks in a *BRCA2*-deficient background.”

2. Statement about accumulation of reversed forks in WRNi-treated *BRCA2*-deficient cells is incomplete too without taking into account that in the absence of *BRCA2* most reversed forks are already degraded and transformed into DSBs. As stated, one might wonder whether WRNi counteracts MUS81-dependent cleavage, which is not the case as shown by the authors.

RESPONSE: We have modified the corresponding text in the Discussion to accommodate the Reviewer’s comment. The revised text found on page 21, lines 631-637 are as follows:
“Consistent with this idea, WRN helicase inhibition resulted in a strong chromatin enrichment of MRE11 that contributes to fork degradation. WRN loss results in fork collapse and MUS81-dependent DSBs following replication stalling²³. Also, in *BRCA2*-deficient cells, partially resected forks are cleaved by MUS81 endonuclease leading to transient accumulation of DSBs³⁵. In agreement with this, we found that WRNi treatment contributes to DSBs in *BRCA2*-mutated cells in a MUS81-dependent manner, suggesting WRNi-induced stalled forks undergo MUS81 cleavage following nascent DNA degradation.”

Thus, I would strongly advise authors to revise the final paragraph of the discussion to reconcile their own data with data from previous works and the first scheme of the model to include previous data from other groups as well.

RESPONSE: We thank the reviewer for the insights he/she provided. We have modified the discussion section to incorporate his/her points. We have also rephrased some wording in the text and cited some previous works that are pertinent to our findings to better reflect the findings in the model.